molecular biology

collagen, liquid chromatography tandem mass spectrometry, mass spectrometry, micro-computed tomography scanning, zooarchaeology, zooarchaeology by mass spectrometry

**Authors for correspondence:**
Theis Zetner Trolle Jensen
e-mail: tztjensen@bio.ku.dk
Hannes Schroeder
e-mail: hschroeder@bio.ku.dk

# The biomolecular characterization of a finger ring contextually dated to the emergence of the Early Neolithic from Syltholm, Denmark

Theis Zetner Trolle Jensen[1,2], Meaghan Mackie[1,3],
Alberto John Taurozzi[1], Liam Thomas Lanigan[1],
Carsten Gundelach[4], Jesper Olsen[5],
Søren Anker Sørensen[6], Matthew James Collins[1,7],
Mikkel Sørensen[8] and Hannes Schroeder[1]

[1]Section for Evolutionary Genomics, The Globe Institute, Faculty of Health, University of Copenhagen, Øster Voldgade 5-7, 1350 Copenhagen, Denmark
[2]BioArCh, Department of Archaeology, Environment Building, Wentworth Way, University of York, York YO10 5NG, UK
[3]Novo Nordisk Foundation Center for Protein Research, University of Copenhagen, Blegdamsvej 3b, 2200 Copenhagen, Denmark.
[4]Department of Physics, NEXMAP, Technical University of Denmark, Fysikvej 311, 2800 Kgs Lyngby, Denmark
[5]Department of Physics and Astronomy, Aarhus University, Ny Munkegade 120, 8000 Aarhus C, Denmark
[6]Museum Lolland-Falster, Frisegade 40, 4800 Nykøbing Falster, Denmark
[7]McDonald Institute for Archaeological Research, University of Cambridge, West Tower, Downing Street, Cambridge CB2 3ER, UK
[8]The Saxo Institute, Department of Archaeology, University of Copenhagen, Karen Blixens vej 4, 2300 København S, Denmark

TZTJ, 0000-0002-7166-7975; MM, 0000-0003-0763-7592;
AJT, 0000-0003-0378-1626; LTL, 0000-0001-6415-3506;
JO, 0000-0002-4445-5520; MJC, 0000-0003-4226-5501;
HS, 0000-0002-6743-0270

We present the analysis of an osseous finger ring from a predominantly early Neolithic context in Denmark. To characterize the artefact and identify the raw material used for its manufacture, we performed micro-computed tomography scanning, zooarchaeology by mass spectrometry (ZooMS) peptide mass fingerprinting, as well as protein sequencing by

liquid chromatography tandem mass spectrometry (LC-MS/MS). We conclude that the ring was made from long bone or antler due to the presence of osteons (Haversian canals). Subsequent ZooMS analysis of collagen I and II indicated that it was made from *Alces alces* or *Cervus elaphus* material. We then used LC-MS/MS analysis to refine our species identification, confirming that the ring was made from *Cervus elaphus*, and to examine the rest of the proteome. This study demonstrates the potential of ancient proteomics for species identification of prehistoric artefacts made from osseous material.

## 1. Introduction

Several excavations at Syltholm near Rødbyhavn on the island of Lolland, Denmark, have revealed an exceptionally well-preserved archaeological assemblage belonging to the Ertebølle (*ca* 7350–5950 cal BP) and Early Funnel Beaker periods (*ca* 5950–4750 cal BP) [1]. Among other things, the assemblage contains numerous artefacts made from organic material, such as wood, bone and antler, as well as several exotic objects, including a T-shaped antler axe [2], a Danubian shaft-hole axe made of amphibolite as well as pieces of Arkadenrand-type ceramics. These finds suggest connections with Neolithic societies of northern Germany and central Europe. One of the more spectacular finds from one Syltholm site (906-II) is one half of an osseous finger ring found in 2014 at the northernmost section of this site (figure 1).

The finger ring (find no. X2784) is broken, but is otherwise perfectly preserved and displays excellent handicraft, design and finish (figure 1*d*). It measures 2.4 cm in diameter, large enough to suggest that it might have been worn by an adult male. The exterior is finely polished, with only microscopic scratches and no use-wear visible, while the interior still shows well-preserved traces of carving, suggesting that it was either barely worn, or that it broke during manufacture. The ring was found in a layer containing a large amount of wooden artefacts, which have been directly dated to between *ca* 6300 and 5500 cal BP (table 1 and figure 2), spanning the period(s) of activity at the site. The ring itself was found close to a broken wooden spear made of ash (X4955, table 1), which yielded a date of 5983–5750 cal BP, and while we were unable to obtain a direct date for the ring itself (due to sampling limitations), we propose that these two contextually associated artefacts are coeval.

Finger rings made of osseous material first appear in large quantities during the Anatolian Neolithic [3], and later over a large area of southern and central Europe. On the Iberian peninsula, numerous finger rings attributed to the Neolithic Cardial culture are known [4–7]. Further north, bone rings are present in deposits from the Rubané culture, e.g. at Mulhouse-Est and in the wider Alsace area (Linear band ceramic (LBK)) [8]. Sporadic occurrences of rings appear from the LBK/Rubané periods and subsequent periods in north and central Europe. In The Netherlands, at Ypenburg 4, a bone ring was found in a child's burial dated to the Middle Neolithic [9]. In northern Germany, rings were found at the sites of Oldenburg Dannau LA 191 and at Wangels, both are dated to the Middle Neolithic (S. Hartz 2018, personal communication). In addition, a limited number of rings dated to the Danish and Swedish Early Neolithic have been found, predominantly in dolmens [10,11]. These artefacts are unlikely to be finger rings due to the large shank depth. The ring from Syltholm is the only example known from the Early Neolithic in Denmark, apart from another broken ring from the shell-midden site at Nederst in Jutland, which was found alongside a ring-preform. Both Nederst artefacts were manufactured from wild boar (*Sus scrofa*) tusk [12], as a thin layer of enamel is visible on the preform surface [12] (E. Kannegaard 2018, personal communication).

The composition of the Syltholm ring is not as readily identifiable as the Nederst ring, and we therefore carried out a series of analyses to identify and characterize the raw material used for its manufacture. X-ray micro-computed tomography (micro-CT) imaging was performed to create high-resolution scans of the ring, while zooarchaeology by mass spectrometry (ZooMS) peptide mass fingerprinting and liquid chromatography tandem mass spectrometry (LC-MS/MS) protein sequencing were used for species identification and further characterization. ZooMS is often chosen for archaeological research because it can provide a rapid, cost effective species identification for samples containing collagen (i.e. bone, antler and skin) [13,14]. However, at present ZooMS is unable to separate the two species of cervids *Cervus elaphus* (red deer, hereafter referred to as *Cervus*) and *Alces alces* (European elk or North American moose, hereafter referred to as *Alces*). Therefore, we also used LC-MS/MS protein sequencing to refine and confirm species identification. We demonstrate the potential of a combined approach for the analysis of prehistoric artefacts made from osseous material and add to the understanding of this hitherto under-studied finds category.

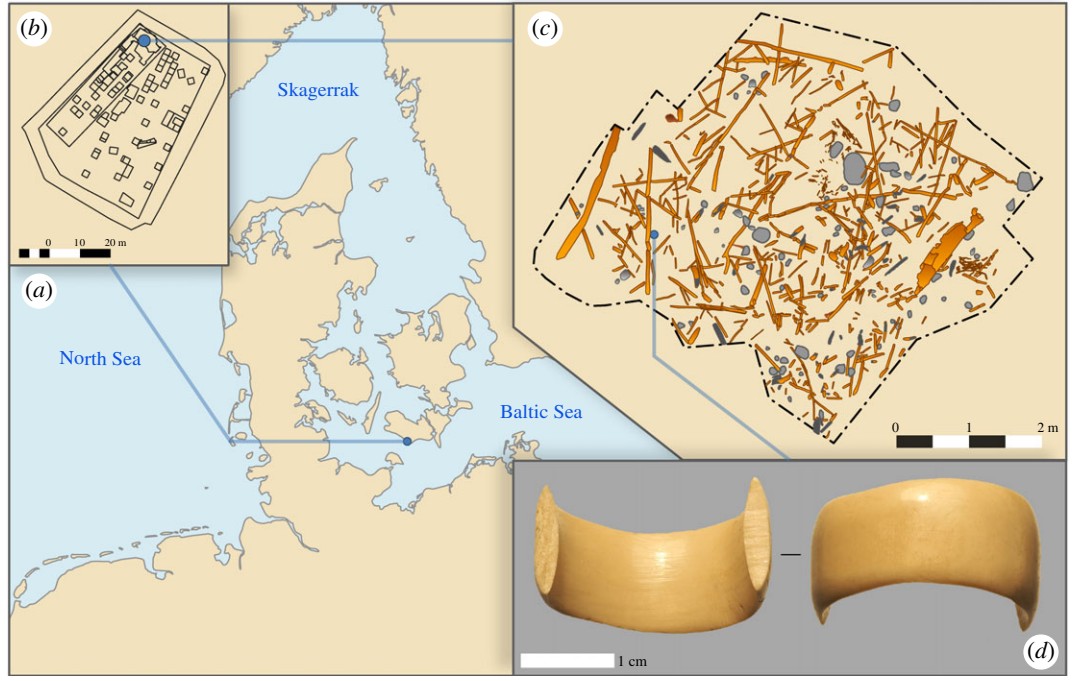

**Figure 1.** (*a*) Location of the site on the southern part of Lolland, Denmark. (*b*) Overview of site MLF906-II where the ring was found in the northern part. (*c*) Digitized archaeological wood and stones found in a small section of the site, from where the ring was found. Digitization based on seven three-dimensional models obtained by Structure from Motion. (*d*) Photograph of the ring.

## 2. Material and methods

### 2.1. The ring in context

Up until 1872 Syltholm was submerged, but after severe flooding on Lolland a reclamation project was undertaken. The area was dammed, thus preserving the inundated Stone Age landscape below. Around 7000 m$^2$ of prehistoric seabed was exposed at the sites MLF906-I and MLF906-II, with the underlying landscape located 1.50–3.00 m below the surface. By the end of the excavation campaign, a total of 80 000 m$^2$ will have been excavated (at all 20 sites) out of *ca* 187 hectares that will become a construction site for the Femern Belt connection [15]. During the Late Mesolithic and Early Neolithic, MLF906-II was located in a shallow brackish lagoon protected from the open sea to the south by shifting sandy barrier islands. Preservation of the site varies according to the degree of shelter provided by the barrier islands; however, the preservation of organic material (wood, bone and antler) is generally very good. Out of the 20 sites excavated to date, the majority of *in situ* finds were uncovered in the reed zone along the banks of the littoral lagoon. The finds were deposited in a coarse brown gyttja varying in thickness (10–50 cm), with no apparent stratigraphy. The layer on top of the gyttja is a clearly defined transgression horizon consisting of sand, shells and vast amounts of water rolled artefacts, suggesting an erosive milieu. Above this layer follows a layer of detritus gyttja, with no archaeological remains, and then a thick layer of sand.

### 2.2. Radiocarbon dating

Direct dating of the ring was not possible due to sampling limitations. However, 70 radiocarbon dates of various artefacts from the site (table 1) were commissioned as part of the wider project carried out by the Museum Lolland-Falster. The radiocarbon measurements were carried out at the Aarhus AMS Centre (AARAMS) at the University of Aarhus, Denmark. Wood samples were pretreated using a standard acid–base–acid procedure (1 M HCl for 1 h at 80°C, followed by 1 M NaOH for 3 h also at 80°C, and finally left overnight in 1 M HCl at room temperature). The pretreatment for bone samples followed a modified Longin procedure [16–18]. Bone minerals were dissolved using 1 M HCl at 4°C for several days followed by removal of humic substances using 0.2 M NaOH, also at 4°C. Subsequently, the extracted collagens were gelatinized in 0.01 M HCl at 58°C overnight. The collagen extracts were

**Table 1.** List of radiocarbon dates. Due to the rarity of the artefact, 15 AMS dates were extracted from the excavations AMS database, from an area surrounding the findspot of the ring, to infer a contemporaneous date of the ring. The material dated is primarily wood, apart from a harpoon made of roe deer (*Capreolus capreolus*) antler.

| finds no. | material | lab no. | $\delta^{13}$C (‰ VPDB) | age ($^{14}$C years BP) | age cal BP (2$\sigma$, modelled) |
|---|---|---|---|---|---|
| X4633 | harpoon, type C (bone) | AAR-24615 | −24.1 | 5412 + 29 | 6288–6183 |
| | boundary (start) | | | | 6165–5960 |
| X8506 | ash spear | AAR-26285 | −27.2 | 5308 + 24 | 6105–5946 |
| X5616 | log boat | AAR-26291 | −28.4 | 5241 + 35 | 6096–5915 |
| X4281 | paddle shaft | AAR-26272 | −27.0 | 5193 + 24 | 5990–5913 |
| X10213 | ash spear | AAR-26283 | −27.0 | 5148 + 26 | 5986–5765 |
| X4955 | ash spear | AAR-24619 | −26.0 | 5128 + 35 | 5983–5750 |
| X8823 | ash spear | AAR-25106 | −27.0 | 5110 + 28 | 5923–5751 |
| X2697 | bow | AAR-22731 | −29.8 | 5045 + 25 | 5896–5730 |
| X10244A | paddle shaft | AAR-26284 | −24.4 | 5041 + 25 | 5895–5724 |
| X8844 | ash spear | AAR-26280 | −26.6 | 5025 + 27 | 5893–5662 |
| X8732 | stake | AAR-25100 | −29.3 | 5024 + 27 | 5892–5662 |
| X8643 | ash spear | AAR-26279 | −25.6 | 5015 + 31 | 5892–5657 |
| X5617 | fish trap | AAR-26271 | −27.6 | 4960 + 30 | 5744–5610 |
| X4282 | ash spear | AAR-26278 | −27.0 | 4909 + 25 | 5710–5595 |
| X4303 | paddle shaft | AAR-26273 | −26.1 | 4887 + 24 | 5655–5593 |
| | boundary (end) | | | | 5649–5512 |

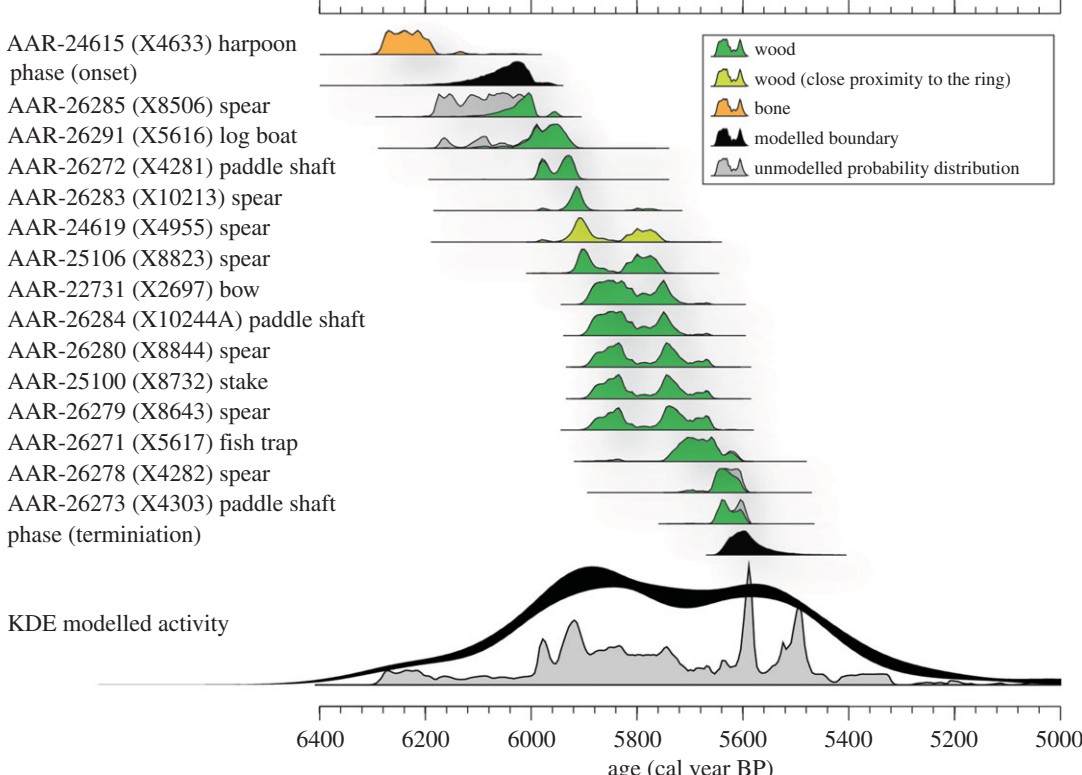

**Figure 2.** Probability distributions of the 15 radiocarbon samples found in close proximity to the ring. The coloured probability distributions are the result of a simple Bayesian model assuming all samples to originate from the same phase of activity. Onset and termination of the phase are indicated with black probability distributions. The light green probability distribution is the date we propose for the ring as well, based on the proximity of the spear to the ring. KDE model of all 70 [14]C dates indicating a single inferred period of archaeological activity at site in general.

ultra-filtered and the resulting greater than 30 kDa collagen fraction was used for radiocarbon analysis. The uncalibrated dates were calibrated using Oxcal v.4.3 and the IntCal13 calibration curve [19,20] and are listed in electronic supplementary material, SI 2 [21].

## 2.3. Imaging/computed tomography scanning

Micro-CT was used to examine the ring using the commercial Zeiss Xradia410 versa system. The ring was rotated 360° in 1601 steps taking a picture at each step using a pre-voltage of 80 kV and a power of 10 W. Two measurements with different pixel resolutions were performed at 32.3 and 13.5 μm. The three-dimensional volume was reconstructed using the software provided with the instrument system 'Reconstructor', which is based on a Feldkamp–Davis–Kress algorithm [22]. The resulting three-dimensional volumes are cylinders with a diameter and height of 3.2 cm and 1.35 cm, respectively, corresponding to the different pixel resolutions, containing different amounts of the object. Visualization was performed using Avizo 9.7 (Thermo Fisher Scientific). The volume investigated with high-resolution X-ray micro-CT has been segmented into elements of the bone (shown in transparent grey) and porosity in the bone (shown in blue). The different levels of blue are a result of the amount of transparency.

## 2.4. Sampling, protein extraction, enzymatic digestion and peptide purification

The artefact surface was decontaminated using 5% bleach followed by 80% ethanol and subsequently 11 mg of bone from one of the fracture planes of the ring was removed using a sterile scalpel. The sampling and protein extraction of the ring were conducted in the dedicated clean laboratories facilities at the Centre for GeoGenetics, University of Copenhagen, Denmark.

To explore the potential for proteomically discriminating between *Cervus* antler and bone in ancient samples, a sample each of both modern *Cervus* bone and antler were collected from the Zoological Museum of Denmark. These samples were taken from a specimen that was defleshed by heating it in

water for 3 days at *ca* 65°C. This experimentally heated extant sample is more comparable to the ring sample than fresh bone or antler. The two reference samples from *Cervus* antler and bone (weighing approx. 15 mg) were subsequently sampled and extracted in a dedicated proteomics laboratories at the Section for Evolutionary Genomics, University of Copenhagen.

The protein extractions were based on a minimally destructive protocol published by van Doorn *et al.* [23] with the following modifications: The samples were incubated in 100 µl of 50 mM $NH_4HCO_3$ (Sigma) for 16 h at ambient temperature. Samples were then agitated using a vortex mixer for 15 s before centrifugation at 13 000 r.p.m. for 1 min, the supernatant was discarded. This step acts as a wash to limit contamination from the burial environment. After, two different extractions were performed for the ring (Extraction 1 and 2), while the reference samples were extracted according to the Extraction 1 protocol. Extraction 2 was performed to remove humic acids that could have contaminated the artefact. Extraction 1: 100 µl of 50 mM $NH_4HCO_3$ was added to the sample before incubation at 65°C for 1 h, the supernatant (Extraction 1) was collected. Extraction 2: the remaining sample was washed three times with 100 µl of 0.1 M NaOH at 4°C and subsequently incubated in 100 µl of 50 mM $NH_4HCO_3$ at 65°C for 1 h, the supernatant (Extraction 2) was collected. Fifty microlitres of each extraction were transferred to a separate 1.5 ml Eppendorf tube, 1 µl of sequencing grade trypsin (0.4 µg µl$^{-1}$) (Promega) was added to each followed by incubation at 37°C for 16 h. After digestion, the extractions were centrifuged at 13 000 r.p.m. for 1 min before acidification to less than pH 2 using 5% (vol/vol) trifluoroacetic acid (TFA, Sigma Aldrich). Purification was performed using C18 reverse phase resin ZipTips (PierceTM) according to the manufacturer's instruction, and the peptides were eluted with 50 µl of 50% acetonitrile (ACN) (Sigma Aldrich)/0.1% TFA (vol/vol).

## 2.5. ZooMS peptide mass fingerprinting

Peptide eluates of the ring were co-crystallized with α-cyano-4-hydroxycinnamic acid (Sigma Aldrich) matrix solution (50% ACN/0.1% TFA (vol/vol)) at a ratio of 1 : 1 (1 µl : 1 µl). Mass spectrometry was performed using a Bruker Ultraflex III (Bruker Daltonics) matrix-assisted laser desorption/ionization time of flight mass spectrometer (MALDI-TOF-MS) run in reflector mode with laser acquisition set to 1200 and acquired over an *m/z* range of 800–3200. The generated spectral output was converted to TXT and was analysed using the open-source software mMass v.5.5.0 [24]. The triplicate raw files were merged, and then peak picked with an S/N threshold of 4. MALDI-TOF-MS was performed at Centre for Excellence in Proteomics at the University of York, UK.

## 2.6. Liquid chromatography tandem mass spectrometry

The leftover peptide eluates of the ring sample were evaporated to dryness using a vacuum concentrator (Eppendorf, Hamburg, Germany), and transferred to the Novo Nordisk Foundation Center for Protein Research, University of Copenhagen for LC-MS/MS analysis on a EASY-nLC 1200 (Proxeon, Odense, Denmark) coupled to a Q Exactive HF-X (Thermo Scientific, Bremen, Germany). The dried peptides from the two ring extractions were resuspended in 100 µl of 80% ACN and 0.1% formic acid (FA), and 15 µl of each of the two ring sample extractions were combined. The combined sample was vacuum centrifuged at 45°C until approximately 3 µl was left, and was then rehydrated with 10 µl of 0.1% TFA, 5% ACN. Protein concentration of elutions was measured by UV absorbance at 205 nm using a Nanodrop (Thermo, Wilmington, DE, USA). The volume required for approximately 2 µg of protein per sample was placed in separate wells on a new 96-well plate and topped up to 30 µl using 40% ACN and 0.1% FA. They were then vacuum centrifuged and resuspended as above, with 5 µl of sample analysed by LC-MS/MS. The LC-MS/MS parameters were the same as previously used for palaeoproteomic samples [25], in short: MS1: 120 k resolution, maximum injection time (IT) 25 ms, scan target 3E6. MS2: 60 k resolution, top 10 mode, maximum IT 118 ms, minimum scan target 3E3, normalized collision energy of 28, dynamic exclusion 20 s, and isolation window of 1.2 *m/z*.

The Thermo RAW files generated were then searched using the software MAXQUANT (v. 1.6.2.6a or v. 1.6.3.4) [26]. The database was prepared using previously published type 1 collagen sequences from *Cervus elaphus* and *Alces alces* [27] (table 2). Missing amino acids in the *Alces* and *Cervus* collagen sequences were substituted with ones from the same positions from the *Bos taurus* sequence (obtained from UniProt, 20-07-18; electronic supplementary material, SI 1). Furthermore, in order to identify proteins in addition to collagen type 1, all protein sequences available for *Cervus elaphus* were downloaded from UniProt (20-07-18). Unfortunately, no other proteins for *Alces* have, thus far, been uploaded to UniProt. MaxQuant settings were as follows. Digestion mode was set to semispecific for Trypsin, to account for possible

**Table 2.** Identified *Cervus elaphus* peptides based on published collagen sequences from [27].

| sequence | length | missed cleave | Da | Q | MQ score | matched spectra |
|---|---|---|---|---|---|---|
| PGEVGPPGPPGPAGEK | 16 | 0 | 1441.7201 | 2 | 218.62 | 11 |
| GETGPAGRPGEVGPPGPPGPAGEK | 24 | 1 | 2167.0658 | 2;3 | 276.46 | 9 |
| PGEVGPPGPPGPAGEKGAPGAD | 22 | 1 | 1909.917 | 2 | 196.48 | 1 |
| GAPGPDGNNGAQGPPGPQGVQGGK | 24 | 0 | 2112.9937 | 2;3 | 319.91 | 9 |
| SGETGASGPPGFAGEK | 16 | 0 | 1447.6579 | 2 | 169.23 | 5 |

additional hydrolytic cleavages occurring during diagenesis. Variable modifications were: oxidation (M), Acetyl (Protein N-term), Deamidation (NQ), Gln → pyro-Glu, Glu → pyro-Glu and Hydroxyproline. Fixed modifications were: Carbamidomethyl (C). The remaining settings were set to the program defaults, apart from Min. score for unmodified and modified peptides searches, which were both set to 60. Proteins were considered confidently identified if at least two razor+unique peptides covering distinct areas of the sequence were recovered (a razor peptide is a peptide which is assigned to the matching protein group with the highest number of peptide identifications and those uniquely assigned to that protein group). MS/MS spectra were assessed manually for confident identification, and peptides from the *Cervus elaphus* Uniprot protein database were searched against the NCBI database using the BLASTp tool (http://blast.ncbi.nlm.nih.gov/Blast.cgi, [28]) to determine species specificity. In addition, the samples were searched against the MaxQuant contaminant database that identifies proteins which may be present due to sample handling and laboratory analysis. Any protein not considered authentic to the ring (i.e. keratins from skin, bovine serum albumin (used as a laboratory standard)) was not included in further analysis except as a comparison for deamidation levels. Deamidation was assessed using publicly available code [25].

# 3. Results

## 3.1. Radiocarbon dates

The 15 calibrated radiocarbon dates, retrieved from close proximity to the ring, revealed an age distribution spanning the Late Ertebølle and the Early Neolithic periods (table 1). A harpoon (X4633) made from roe deer (*Capreolus capreolus*) antler produced the oldest date. The date of this artefact is in good agreement with directly and indirectly dated specimens of similar typology and raw material [29]. The collagen yield of the harpoon sample was 5.1% and it yielded stable isotope values ($\delta^{13}$C = −24.1‰, $\delta^{15}$N = 4.4‰ and C/N = 3.3) consistent with already published archaeological roe deer values in Denmark [30]. The remaining dated material was not typologically dated to a specific cultural period. The dates demonstrate that the site was frequented both before and after the Mesolithic/Neolithic transition, indicating that the site was continuously occupied during this transitional period. The usage period of the site was estimated using a simple Bayesian model assuming all finds to originate from a single phase of activity (figure 2). The onset of activity at the site is estimated to be *ca* 6060 cal BP (6173–5962 2$\sigma$) and the activity is suggested to end *ca* 5590 cal BP (5649–5512 2$\sigma$). The harpoon was excluded from the model due a low statistical agreement in the Bayesian model [17]. The reason for this is unknown as all collagen quality parameters are within expected ranges; however, it is possible that the harpoon was redeposited. The ring dates to the main occupation period of the site (*ca* 6060 cal BP–*ca* 5590 cal BP). The ash (*Fraxinus excelsior*) spear found closest to the ring returned a date of 5983–5750 cal BP (5128 ± 35 [14]C yr BP). Due to their close contextual and stratigraphic association, we propose that the ring has a similar date, but we cannot rule out that the ring is Late Mesolithic. However, as shown in the kernel density estimation (KDE) model, which includes an additional 55 dates from the surrounding area, activity at the site in general peaks during the Early Neolithic (figure 2).

## 3.2. Imaging/computed tomography scanning

Micro-CT imaging revealed the presence of many evenly spaced circular pores (figure 3a–d). The frequency and morphology of these pores indicate that they are osteons or Haversian canals.

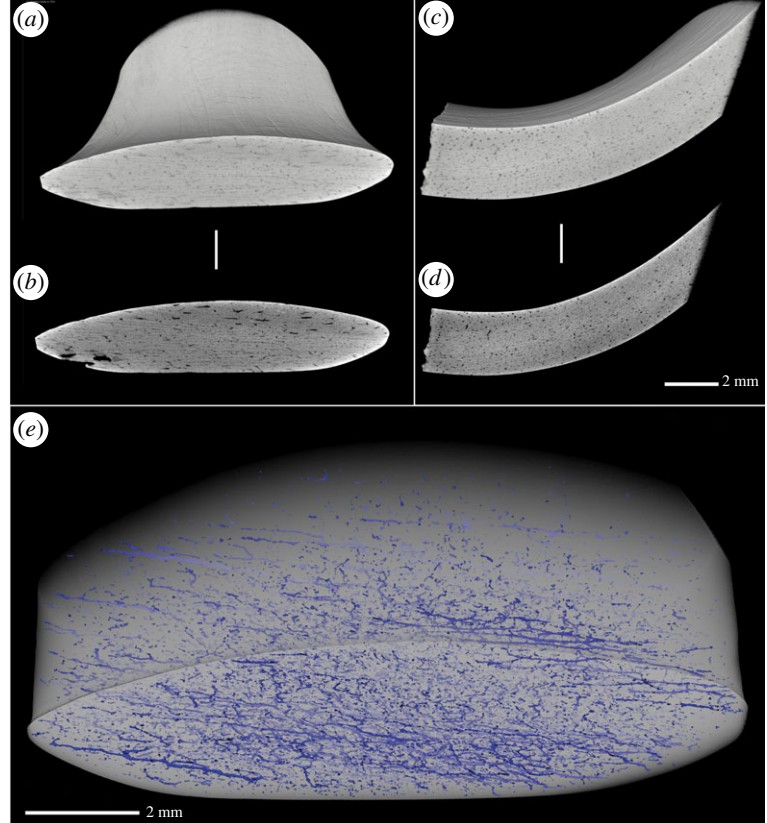

**Figure 3.** Micro-CT scan. (*a*) Full field of view of volume rendering, (*b*) transverse slice showing a few small black pores assumed to be Volkmann's canals, (*c*) cut along the middle of the volume rendering, (*d*) slice along showing several small black pores assumed to be osteons, (*e*) network of osteons arranged longitudinally and Volkmann's canals aligned perpendicular to the latter (in blue).

Additionally, Volkmann's canals, which are perpendicular connections between Haversian canals [31], can be seen in figure 3*e* and in the rendered supplementary animation (electronic supplementary material, video S1). The diameter of osteons in antler ranges from 100 to 225 µm, similar to those found in bovine femur [32]. By contrast, enamel only contains nanopores, and tubules in dentine and ivory (mostly composed of dentine) are approximately 1–2 µm in diameter [33]. We therefore conclude that the ring was manufactured from antler or bone, rather than enamel or dentine. Unfortunately, distinguishing archaeological antler from bone, where only the compact bone is present, using micro-CT scanning is problematic and inconsistent [34]. This is because the identification is based on subtle size differences in the diameter of the osteons, which are affected by diagenesis [34,35].

## 3.3. ZooMS peptide mass fingerprinting

To identify the species used to manufacture the Syltholm ring, we performed ZooMS peptide mass fingerprinting by MALDI-TOF-MS on the two extracts. The spectral outputs revealed a series of isotope distributions corresponding to the mass of tryptic peptide products within a range of 1105–3101 Da. Peptide masses previously reported to be unique for *Cervus* and *Alces* collagen 1 (COL1) were observed with high intensity [27,36] (figure 4). However, these two closely related species cannot be separated at present using ZooMS.

The two extraction methods showed some differences in terms of peptide count as well as intensity. Extraction 1 showed greater intensity peaks of lower molecular weight peptides than extraction 2. However, spectral data from extraction 2 showed greater intensity of a large portion of the higher molecular weight peptides. The reason for this small difference remains unknown. Given the spectral similarity between the two extractions, it is likely that the artefact was not contaminated with humic acids from the environment (figure 4).

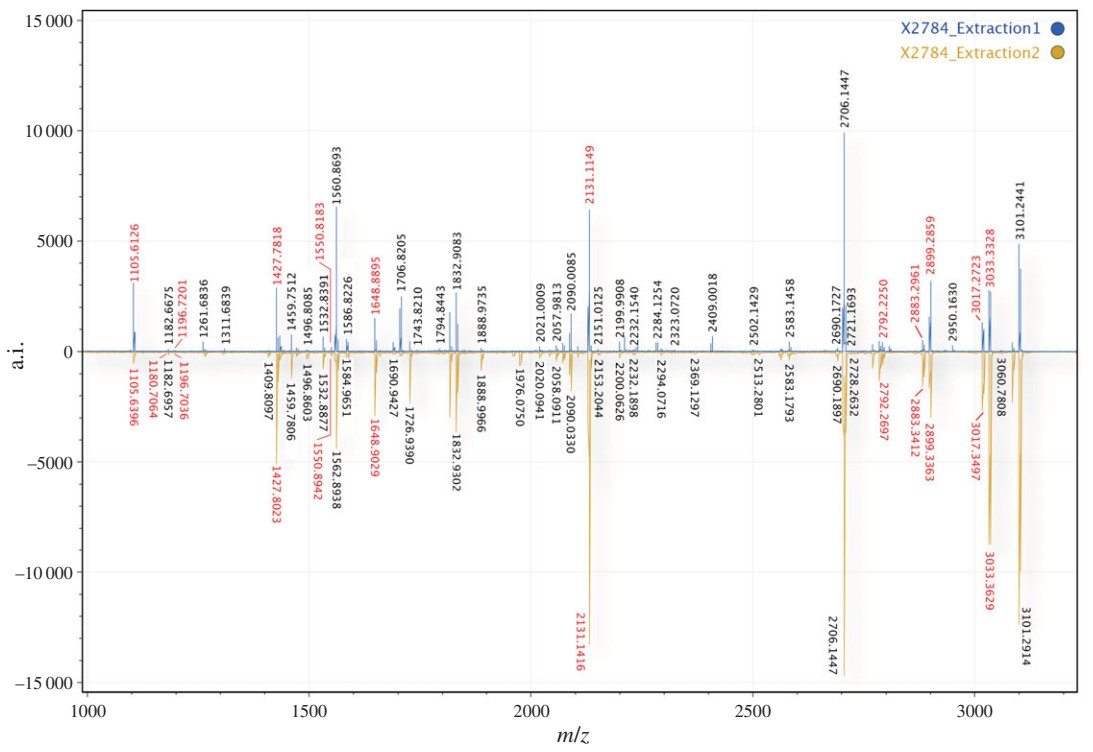

**Figure 4.** ZooMS results. MALDI-TOF-MS spectral output visualized in mMass (v.5.5.0) Extraction 1 and 2 flipped. Peptides unique to *Cervus* and *Alces* are highlighted in red.

## 3.4. Cervidae species identification by Liquid chromatography tandem mass spectrometry

In order to differentiate between *Cervus* and *Alces*, we performed LC-MS/MS analysis of the combined protein extracts. While the collagen type 1 (α1 and α2) chains in *Cervus* and *Alces* are highly conserved, three known SAPs (single amino acid polymorphisms) exist and can be used to separate these species. The sites are position 741 P (*Cervus elaphus*) or A (*Alces alces*) on the α1 chain, and 454 P (*Cervus elaphus*) or I/L (*Alces alces*), and 749 S (*Cervus elaphus*) or T (*Alces alces*) on the α2 chain. Using a MaxQuant search of our custom database, we identified peptides that map uniquely to the *Cervus* sequence at all three positions (two of which are shown in figure 5 as well as in table 2).

## 3.5. Additional proteins detected in the Syltholm ring

In addition to COL1, we identified 14 other endogenous proteins using LC-MS/MS excluding contaminants (electronic supplementary material, SI 3). These were additional collagens (collagen type 3 α1 (COL3A1), collagen type 11 (COL11A2) and collagen type 12 α1 (COL12A1)), blood proteins, such as immunoglobulin gamma-1 heavy chain (IGHG1), serum albumin (ALB), apolipoprotein A-I (APOA1), and nucleobindin 1 (NUCB1), and additional extracellular matrix proteins, such as osteocalcin (BGLAP), alpha 2-HS glycoprotein (AHSG), pigment epithelium-derived factor/serpin family F member 1 (SERPINF1), thrombospondin 1 (THBS1), biglycan (BGN), secreted phosphoprotein 2 (SPP2) and periostin (POSTN). In some cases, these proteins were identified specifically to *Cervus* (AHSG, IGHG1) or *Cervus/Odocoileus virginianus* (white-tailed deer) (SERPINF1, APOA1) when compared to all publicly available sequences for these proteins (which do not include sequences specific to *Alces*). Since white-tailed deer are native to the Americas, they can be excluded from species identification in this context. However, since the corresponding *Alces* sequences are not available, it cannot be ruled out that these may also match this species, and therefore the results reflect current data availability. For the other identified proteins, we recovered peptides that were less species-specific (electronic supplementary material, SI 3).

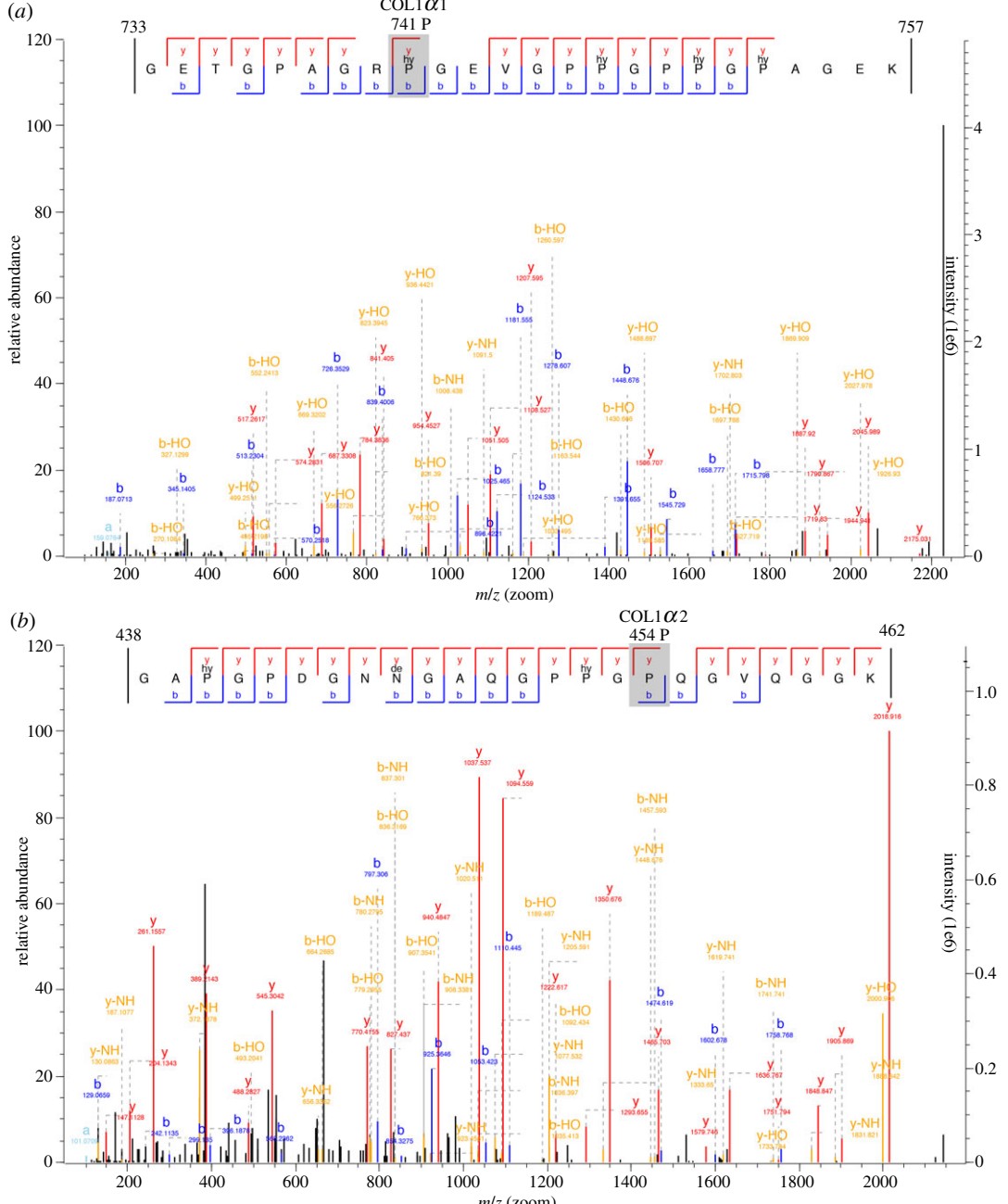

**Figure 5.** Example of two peptides (a,b) from the MS/MS output from MaxQuant, containing SAPs unique for *Cervus* (marked in grey). Panel (a) is located on the collagen 1 α-1 sequence, while (b) is located on the collagen 1 α-2 and both can confidently, based on the y and b ion series, be identified as *Cervus*.

## 3.6. Modern bone and antler proteomes

The experimentally heated extant bone and antler samples yielded 18 and 29 proteins, respectively (electronic supplementary material, SI 4). Even though they were modern samples, the same requirement of at least two razor+unique peptides for identification was followed. Both samples unsurprisingly contained collagens, namely COL1, COL3A1 and collagen type 5 α1 (COL5A1). Blood proteins were highly represented, especially in the antler. AHSG, SERPINF1, ALB, NUCB1 and tetronectin (CLEC3B) were found in both the antler and bone, whereas haemoglobin (HBB), antithrombin-III/serpin family C member 1 (SERPINC1), vitronectin (VTN), immunoglobulin lambda-1 light chain (IGL), serpin family D member 1 (SERPIND1) and APOA1 were found in the antler sample only. In addition, at least nine extracellular proteins were found in both the bone and antler samples: lumican (LUM), decorin (DCN), chondroadherin (CHAD), olfactomedin-like protein 1

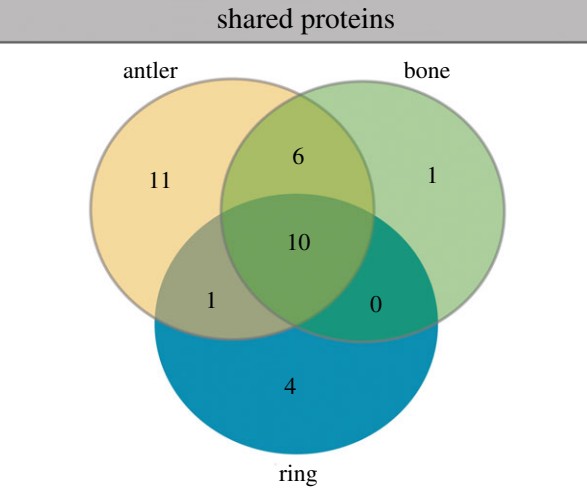

| samples | total | proteins |
|---|---|---|
| antler/bone/ring | 10 | BGN, ALB, BGLAP, COL3A1, COL1, NUCB1, THBS1, SPP2, SERPINF1, AHSG |
| antler/bone | 6 | ANXA2, DCN, CLEC3B, OLFML 1, LUM, CHAD |
| antler/ring | 1 | APOA1 |
| antler | 11 | SPARC, COL5A1, HBB, SERPINC1, SERPIND1, IGL, VTN, S100A12, PCOLCE, S100A8, PGLYRP1 |
| bone | 1 | COL5A2 |
| ring | 4 | COL12A1, IGHG1, COL11A2, POSTN |

**Figure 6.** Venn diagram demonstrating 1 protein shared between the ring and antler, and 0 proteins shared between the ring and bone.

(OLFML1), annexin A2 (ANXA2), BGN, BGLAP, SPP2 and THBS1. The antler sample also contained evidence of five other extracellular proteins: S100 calcium-binding proteins A8 and A12 (S100A8, S100A12), osteonectin (SPARC), procollagen C-endopeptidase enhancer (PCOLCE) and peptidoglycan-recognition protein (PGLYRP1). In general, there is significant overlap in the proteomes of all three samples, i.e. 10 proteins present in all samples. The sample with the most unique proteins is the antler (11 proteins), followed by the ring (four proteins) and the bone sample has only a single unique protein. The plasma protein APOA1 was uniquely recovered in our extractions of the ring and the heated modern *Cervus* antler. This is also visualized in a Venn diagram (figure 6). No unique proteins were recovered between only the ring and heated modern *Cervus* bone.

## 3.7. Liquid chromatography tandem mass spectrometry protein authentication

To assess the authenticity of the proteins recovered, we examined the deamidation patterns of asparagine (Asn/N) and glutamine (Gln/Q). Here, we observed a much higher deamidation rate in the ring sample. On average, the ring sample expressed 57.5% (SD 3.3%) Asn and 26.1% (SD 1.9%) Gln deamidation, while the contaminants showed 5.7% (SD 2.3%) and 0.6% (SD 0.6%), respectively (figure 7). This damage process occurs naturally over time, and while confounded by chemical and environmental factors (such as pH, temperature and humidity [37,38]), these results indicate that the proteins examined in our analysis are likely not preserved well enough to be modern contamination, due to the observed greater amount of damage detected when compared to known contaminant proteins. Additionally, we examined deamidation of the heated modern reference samples, which also have considerably less deamidation than the bone ring. The levels of deamidation present in the modern antler and bone are mostly of the faster reacting asparagine, and may have been caused by the heating step of the defleshing process they underwent before becoming a part of the museum's reference collection (figure 7).

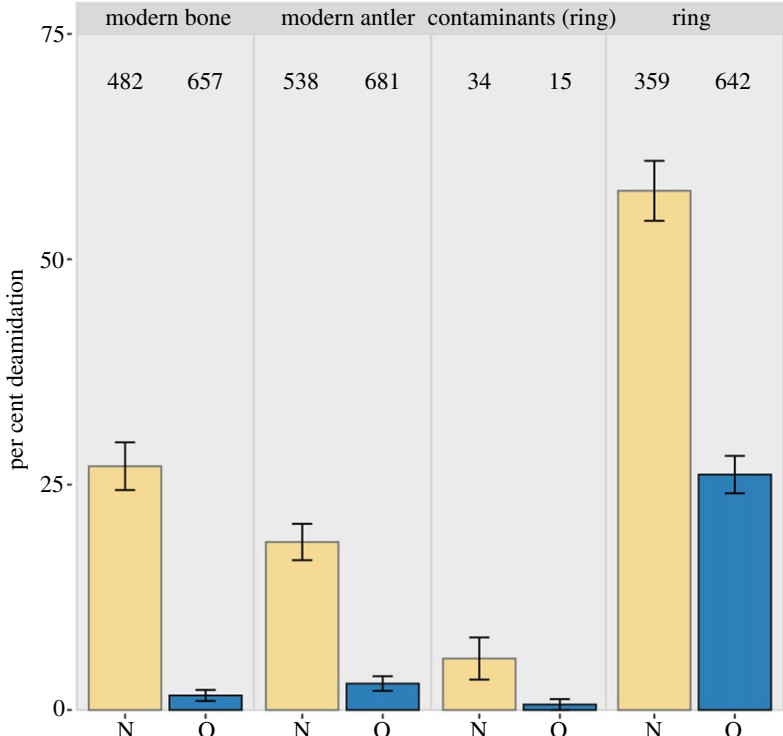

**Figure 7.** Deamidation comparison of the ring, heated modern reference samples as well as contaminant proteins. Numbers in each column denote number of peptides used for the calculation. Deamidated asparagine (N) and glutamine (Q) residues from the ring and known contaminant proteins, showing a considerably higher deamidation rate in the sample, which is evidence of authentically ancient proteins.

# 4. Discussion

Skeletal fragments and heavily worked artefacts often lack morphological osteological landmarks, and are nearly impossible to identify to the species level, let alone to skeletal element using osteological analysis. With the increased availability of advanced analytical techniques, molecular level resolution can be used to answer archaeological questions such as resource exploitation, manufacturing technology and trade.

## 4.1. The Syltholm finger ring is made from *Cervus* antler or bone

Bayesian modelling of 15 radiocarbon dates, obtained from the immediate proximity to the ring was used to indirectly date the ring to the Early Neolithic. Additionally, KDE modelling using a total of 70 dates confirmed that while the site was frequented in the Mesolithic the most intense activity was in the Early Neolithic. Micro-CT was employed to determine the skeletal element used to make the ring. Enamel and bone/antler are easily distinguished from one another by non-destructive micro-CT through visualization of Haversian canals, which are absent in enamel and dentine (including ivory) [35]. The scans (figure 3) clearly show the presence of osteons; therefore, we can exclude enamel/dentine as the raw material.

Having identified the material as deriving from bone or antler, ZooMS was performed to obtain species identification. The collagen I and II peptide mass fingerprint revealed that the ring was manufactured from either *Cervus* or *Alces*. At present, it is not possible to discriminate between some closely related species such as these, using ZooMS, due to the conserved nature of collagen I and II. A positive identification of *Alces* would imply that either an *Alces* bone/antler or the ring itself was imported. However, given the choice between the two species indicated by ZooMS, *Cervus* would be the most likely candidate. *Alces* disappeared at some point in time in this area due to rising sea-levels (Littorina transgressions (starting *ca* 8400 BP)) that effectively turned Denmark into an archipelago, whereas *Cervus* was still abundant in Denmark [39] at that time. *Alces* material could have been introduced to the site through trade; however, no evidence of this species has been recovered at

Sylthom from this period. LC-MS/MS protein sequencing was used to refine our ZooMS species identification. Three peptides with SAPs specific to *Cervus* were observed, confirming it as the source species.

## 4.2. Antler or bone?

Having identified the species from which the ring was crafted, we aimed to discern the skeletal element used, in an attempt to gather more information about the manufacturing process. As mentioned above, it is generally not possible to differentiate between archaeological antler and bone using micro-CT scanning due to bone diagenesis. Morphologically, it is more likely that antler was used, as the cross section of a mature antler tine is approximately the right size and shape of the Syltholm ring, aiding the manufacturing process. By contrast long bone is less circular, necessitating additional work to achieve a ring shape. Antler is also less energy consuming to attain, since it is shed yearly and does not require hunting an animal. Therefore, without biomolecular analysis, antler could be the most likely originating tissue for the ring.

At present, there has not been enough proteomic analysis of the differences in protein presence and expression levels between antler and bone. The two tissues are very similar and most variance could be down to quantitative differences between proteins instead of simply presence or absence. We attempted to investigate this further by generating 'reference' proteomes of *Cervus* antler and bone to which the ring could be compared (electronic supplementary material, SI 4). Due to the small proteome recovered from the ring, it would be inappropriate to compare this dataset against a modern proteome. To this end, we selected a *Cervus* specimen that had been experimentally heated [40] and performed the same extraction protocol that was used for the ring sample.

Unfortunately, comparison of the ring proteome versus the antler and bone did not enable us to confidently assign the ring to either. Due to the limitations of this study, we could not perform a quantitative analysis, partly due to methodology and partly due to the limited recovery of proteins of interest. To our knowledge, there has been no quantitative proteomic comparison between these two tissues, neither with modern nor archaeological samples [41]. Stéger *et al*. [41], however, did quantitatively examine modern *Cervus elaphus* antler and bone for differences in gene expression. They found that the expression of eight genes were 10–30-fold times more expressed in the ossified portion of antler than in skeletal bone from the same individual. Four of these proteins (COL1, COL3, BGLAP and SPARC) were also found in the ring sample. These are all proteins associated with skeletal development, and can be found in both antler and bone. However, it is unsurprising that bone development proteins are more abundant in antler as it is the fastest growing mammalian tissue [42] due to yearly regeneration. While our analyses were not quantitative, it is not out of the question that more abundant proteins would be more likely to be recovered, especially from a proteome depleted sample (due to taphonomic processes). Other proteins recovered from the Syltholm ring are associated with bone formation and mineralization, such as: AHSG [43], SPP2 [44] and POSTN [45]. Collagens type 3, 11 and 12 were recovered, all of which are associated with collagen formation in growing bone [46,47] and present in the growing antler [48,49]. Normally, COL11 and 12 are associated with cartilage, not mineralized bone [47,50,51], but are considered abundant in antler [48,49]. COL11 and COL12 have been recovered from archaeological bone (e.g. [52,53]), but Sawafuji *et al.* [53] have shown that the protein abundance score of COL12 decreases in older human individuals compared to those younger, correllating with the relative amount of bone growth.

We also discovered multiple blood proteins in the ring (IGHG1, ALB, APOA1 and NUCB1). Growing antlers are a highly vascular tissue and contain at least twice as much blood at their peak growth as ovine rib bones, which decreases as the antler ossifies [54]. APOA1, a major component of plasma high-density lipoprotein shown to be linked to osteoblastgenesis and bone synthesis [55], was the only protein uniquely recovered between the modern antler and the ring, although it has also been found in archaeological bone [52,53] and could represent missed recovery in the bone sample. It is of note that APOA1 was only detected by Sawafuji *et al*. [53] in the infant remains studied [53], indicating association with bone formation. It, along with SERPINF1, POSTN and THBS1, has been implicated in axon/nerve growth in growing antlers [56]. SERPINF1 and THBS1 have also been implicated in the stimulation and remodelling of vasculature in antler cartilage, respectively [56,57]. However, these proteins are also recovered from bone samples, albeit highly associated with bone growth and remodelling: SERPINF1 being expressed by osteoblasts during endochondrial bone formation [58], POSTIN is highly expressed in the periosteum and highly active during bone growth and contributes

to changes in bone diameter and cortical thickness [45], and THBS1 is implicated in the remodelling of bone, maintenance of bone mass and fracture healing [59,60].

Therefore, we suggest that the proteins recovered, especially those related to increased bone growth and high vascularization, are consistent with a possible tissue identification of antler, suggested based on the ease of manufacture of this item from antler. However, there is not sufficient evidence to rule out bone either. We greatly encourage more research be undertaken to confirm the proteomic differences between antler and skeletal bone, as it would be valuable for future palaeoproteomic studies and the understanding of archaeological worked ossified objects.

## 4.3. Species identification sheds light on resource exploitation and manufacturing processes

The only coeval ring we know of from the Danish area is the ring from the Nederst shell-midden, on the Djursland peninsula of Jutland (approx. 200 km away). The Nederst ring is considerably smaller (Ø = 1.5 cm) and thus may have been made to fit an adult female or a child. As mentioned earlier, Kannegaard [12] suggests that the raw material used for manufacture was wild boar canine. From the same site, a small rectangular enamel/dentine disc was also found, with a semicircle removed by carving [12], thus showing that manufacturing took place at the midden. At another shell-midden, Stubbe Station, Jutland, two wild boar tusks with transverse saw marks were found. One shows a hole drilled into it, which seems to have cracked the tusk longitudinally [61]. A possible manufacturing process (or *chaîne opératoire*) of dentine/enamel rings can be suggested from these finds: (i) extracting the tusk of a wild boar, (ii) drilling a hole of desired size into the tusk, (iii) transverse sawing on either side to liberate a rectangular preform, (iv) removing the corners of the rectangle and making it circular, (v) grinding and polishing of the exterior. Since no waste material has so far been found from osseous finger ring production in Syltholm, its *chaîne opératoire* cannot be established at present. However, one can imagine that production of such a ring would require: (i) obtaining an antler or bone of a *Cervus*, (ii) transverse sawing to obtain a rough-out, (iii) scraping off the velvet bone for antler; more intensive circular shaping for bone, (iv) removal of the interior trabecular tissue using a flint borer and (v) polishing of the exterior.

Apart from being broken, the ring was well preserved and does not show any evidence of use-wear, which suggests that it was produced in the vicinity of the site. Based on the limited evidence presented here, it may be the case that there were geographically distinct *chaîne operatoires* for ring production during the transition period; the eastern part favouring rings made from antler tines or bone, whereas the western part favoured tusks. While it cannot be ruled out that the Syltholm ring was not manufactured at the site (trade item), there is evidence of regional differences in manufacturing techniques and resource exploitation in Denmark on both sides of the transition. During the Ertebølle period, bone combs, large circular bone rings cut from the scapulae of aurochs (*Bos primigenius*), extinct on Zealand during this period, are almost exclusively found in the western part of Denmark [62], whereas Limhamn and Oringe axes and adzes are mostly found in the eastern part and in Scania [63,64]. During the Early Neolithic, Volling-type ceramics are only found in the western part, whereas Svaleklint ceramics cluster in eastern areas [65].

# 5. Conclusion

This study presents the analysis of a unique artefact, an osseous finger ring, from the period of the Neolithization of northern Europe, contextually dated to the cusp of the Neolithic. We demonstrate the potential of combining several analytical methods on highly worked archaeological osseous artefacts to obtain a plethora of information even from small quantities of sample. The tomographic analysis revealed a matrix of osteons thus assigning the skeletal element to bone/antler, not enamel/ dentine as used for the coveal and proximal Nederst ring. Peptide mass fingerprinting and protein sequencing revealed that the species used to manufacture the Syltholm ring was *Cervus elaphus*. Unfortunately, we were not able to determine if the ring was made of bone or antler. Nonetheless, this study demonstrates how ancient proteomics can still help identify and characterize the source of osseous material used in the manufacture of artefacts, which in turn can be used to infer regional differences in manufacturing processes and resource procurement.

Data accessibility. The mass spectrometry data for both ZooMS and LC-MS/MS have been deposited on the PRIDE Archive [66] with the dataset identifier PXD011811.

Authors' contributions. T.Z.T.J. and S.A.S. initiated the study. S.A.S. provided samples and contextual information. C.G. and T.Z.T.J. conducted the micro-CT scan and interpretations. Radiocarbon dates were produced and interpreted by J.O. and T.Z.T.J. T.Z.T.J. performed the ZooMS experiments. T.Z.T.J., M.M., A.J.T. and L.T.L. collected, prepared, and ran samples for MALDI-TOF-MS and LC-MS/MS. T.Z.T.J., M.M., A.J.T., L.T.L. and H.S. interpreted data and wrote the paper, with critical input from M.J.C. and M.S. All authors reviewed and approved the manuscript.

Competing interests. We declare we have no competing interests.

Funding. T.Z.T.J. is supported by the European Union's EU Framework Programme for Research and Innovation Horizon 2020 under grant agreement no. 676154 (ArchSci2020). M.M., A.J.T., L.T.L. and M.J.C. are funded by the Danish National Research Foundation award PROTEIOS (DNRF128). H.S. was supported, in part, by the Villum Foundation (grant no. 22917) and the Carlsberg Foundation (grant no. CF19-0601). Work at the Novo Nordisk Foundation Center for Protein Research is funded in part by a donation from the Novo Nordisk Foundation (grant no. NNF14CC0001).

Acknowledgements. We thank the following for valuable discussion: Dr Lutz Klassen and Esben Kannegaard (Museum Østjylland), Dr Sönke Hartz (Archäologisches Landesmuseum), Dr Selena Vitezović (University of Belgrade) and Bharath Nair (University of Copenhagen). The following are greatly acknowledged: Museum Lolland-Falster for sample access, Centre of Excellence in Mass Spectrometry, University of York for access to MALDI-TOF-MS and The 3D Imaging Centre at The Technical University of Denmark for access to micro-CT. Prof. Jesper Velgaard Olsen at the Novo Nordisk Center for Protein Research for providing access and resources. The authors would also like to thank Kristian Murphy Gregersen of the Zoological Museum of Denmark for providing the modern *Cervus* comparative samples.

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
