## [Reviewer comments · Royal Society Open Science]

Review History

RSOS-191172.R0 (Original submission)

Review form: Reviewer 1 (John Meadows)

Is the manuscript scientifically sound in its present form?

Yes

Are the interpretations and conclusions justified by the results?

Yes

Is the language acceptable?

No

Do you have any ethical concerns with this paper?

No

Have you any concerns about statistical analyses in this paper?

No

Recommendation?

Accept with minor revision (please list in comments)

Comments to the Author(s)

The manuscript is excellent but wordy; try to avoid repetition and redundant information. The following specific suggestions may improve the paper.

Abstract:

if you are confident that your results identify the raw material as antler rather than bone (e.g. p.10 line 59), I would mention this in the abstract.

Use-wear (e.g. p.10 line 33):

why would you expect to see use-wear on the internal surface of the ring, even if it was used? Is the ring in a more pristine condition than those discussed in para.3 of the introduction? If the ring was in pristine condition, indicating that it might have broken during manufacture (p.2 line 20), why not mention absence of use-wear on the external (polished) surface?

Micro-CT:

I was unable to open the supplementary video – perhaps the format is too restrictive. The results appear to exclude ivory as well as enamel or dentine – this should probably be mentioned in the discussion (e.g. p.8 line 37).

Dating:

I am not convinced that the broken ring fragment is necessarily the same date as the closest wooden artefact dated (p.2 line 26, p.6 line 12, Fig. 2). No evidence has been presented except proximity, but the taphonomic situation and the fact that both finds are broken and incomplete means that there's a high risk that their proximity is coincidental. The ring, like the harpoon fragment in the same deposit, might even be Mesolithic.

A better way to estimate the date of any undated find from this deposit is to use the scatter of dates on all artefactual material to derive a single date range. The easiest method is to add one line of code to the OxCal model, within the bounded phase:

Date ("ring");

This function (see Bronk Ramsey 2009, your ref.19) generates a probability density function for the date of an event between the onset and termination boundaries you have calculated based on the spread of dates from wooden artefacts. It would still be misleading if the ring was re-deposited from older layers, but if it was freshly deposited in the gyttja during the period that wooden spears etc were discarded at this location, it would be a more realistic estimate than just picking one spear's date.

The dating results section is contradictory: "The harpoon was excluded from the model due [to] a low statistical agreement in the Bayesian model (see 17). The reason for this is unknown as all collagen quality parameters are within expected ranges." The reason is obvious; you got the right date on the harpoon (given the Late Mesolithic typological dating mentioned earlier in the paragraph) but the wooden artefacts belong to a later (early Neolithic) phase of occupation. There are plenty of ways in which a fragment of an older harpoon could have been deposited at this location during the early Neolithic. The problem is therefore why you should be confident that the ring fragment was freshly deposited, and that it belongs to the same phase of activity as the wooden artefacts.

Also, "The dates demonstrate that the site was frequented both before and after the Mesolithic/Neolithic transition" is true (assuming a transition at c.4000 cal BC, which should have been indicated in para.1 of the introduction), but "The usage period of the site was

estimated using a simple Bayesian model ... The onset of activity at the site is estimated to be c. 6060 cal. BP" is misleading, in the sense that this model applies only to the 15 wooden artefacts at this specific location, and not to Syltholm in general. Simply relying on these 15 dates I would not be confident that there was a Mesolithic phase at Syltholm at all.

LC-MS/MS methods

Some clarification may be required here. The fact that "At present, no proteins for elk have been uploaded to UniProt" seems to imply that you could not actually exclude elk, yet in the results (p.6 line 54) you talk about 3 known SAPs which can be used to separate deer from elk. Where did the elk reference data come from to support this claim? Incidentally, it is impossible for a non-specialist to see how Fig 5 shows the results indicated in the text.

Figure 6 caption "Venn diagram demonstrating 5 proteins shared between the ring and antler, and 0 proteins shared between the ring and bone", but figure shows 10 proteins shared by all 3 samples, and only 1 shared between the ring and antler (but not the bone).

LC-MS/MS protein authentication

What are contaminant proteins, and how do you know that they are contaminants? I can see that from the low deamidation rate (Figure 7) they should not be from the ring itself, but where does this contamination come from?

Minor text edits:

p.2 line 8: do not use "bp" for dates (in the 1980s it was used in archaeological literature for uncalibrated radiocarbon ages). Here you mean "cal BP" (as applied in the rest of the paper), which would be correct. However, Neolithic archaeology in Denmark tends to use the cal BC scale, and unless you feel strongly that this practice should change, I would convert all the calibrated dates into cal BC, including in Fig. 2.

p.2 line 10: "exotic objects, including a T-shaped antler axe" – T-axes are quintessentially Ertebølle artefacts, so I don't see why you regard this one as exotic

p.2 lines 11-12: "... Arkadenrand and Stichband type ceramics. These finds suggest connections with Neolithic societies of northern Germany" – perhaps superfluous information here, but the former would date to the mid-4th millennium cal BC while the latter is much older (early 5th millennium cal BC). Since there is no contextual association between these imports and the osseous ring, I am not sure it's relevant, but perhaps insert "of the 4th and 5th millennia cal BC" after "northern Germany". The confusing aspect for readers is that at Syltholm the 5th millennium is still the Mesolithic, as alluded to in the next paragraph. I think it would be clearer if you just delete the text "This indicates that .. and the beginning of the Funnel Beaker Culture." (p.2 lines 23-24) and insert Mesolithic and Neolithic in the first sentence of para.1, i.e. "...belonging to the Mesolithic Ertebølle... and Early Neolithic Funnel Beaker periods".

The third paragraph of the introduction gives many potential parallels for the ring elsewhere in Europe, but no absolute dates. Non-European archaeologists (and other readers) would probably not realise that Anatolian Neolithic could be much earlier, that LBK and Rubané are the same thing and date to the late 6th millennium cal BC, that Middle Neolithic in the examples given is probably 3rd millennium, etc.

p.3 lines 10-16: I found this description confusing (assuming that I was unfamiliar with the site and the project, and could not read your ref.15, which is in Danish); much of this information is probably redundant for the purpose of this paper.

p.4 line 28: "They were then vortexed" – laboratory colloquialism!

p.5 line 26: “all proteins available for red deer were downloaded” – another colloquialism; data were downloaded or uploaded, not proteins.

“searched”: there are several instances (e.g. in “The Thermo RAW files generated were then searched using the software MaxQuant”) in which this term (or similar expressions such as “peak picked”) are used colloquially or ungrammatically. I think you mean that you used software X to look for reference spectra closely matching the data files.

p.5 line 48: “Groupings were generated by applying kmeans clustering (number of clusters = 3)” – Hyphenate k-means. Groupings of what? Why 3 clusters?

p.8, line 45ff: These sentences “However, the most likely candidate, based on the ZooMS data, would be red deer since elk was part of the depauperate fauna (39). Red deer were still abundant in Denmark during the Atlantic period, as opposed to elk, which disappeared following the Littorina transgressions that turned Denmark into an archipelago.” should be re-written, because (1) the argument is not based on the ZooMS results, but on biogeography; (2) “depauperate” is not a common word (“impoverished” would be more widely understood) and the implicit point is that elk was not part of the impoverished Danish fauna; (3) the Littorina transgression is not dated; (4) the Atlantic period is not dated (I know what you mean, but many readers will not, and might assume that the Atlantic period was initiated by the Littorina transgression). A much simpler chain of reasoning might be that there is abundant red deer bone at Syltholm but no elk bones have been identified, if you know this to be the case.

Table 1 and Figure 2: what is a “fishing ruse”?

Review form: Reviewer 2

Is the manuscript scientifically sound in its present form?

Yes

Are the interpretations and conclusions justified by the results?

No

Is the language acceptable?

Yes

Do you have any ethical concerns with this paper?

No

Have you any concerns about statistical analyses in this paper?

No

Recommendation?

Major revision is needed (please make suggestions in comments)

Comments to the Author(s)

In this manuscript, Jensen et al. attempt to determine the species and skeletal element of origin of an osseous ring from Denmark. They perform μ CT, ZooMS, and LC-MS to evaluate it. This is an interesting study, but has some major interpretation issues that make its conclusions unconvincing. Aspects of the experimental setup also limit the conclusions as well.

One minor thing to revise throughout, for clarity throughout please refer to red deer and elk by their scientific names only. Elk has a very different meaning in North America (where they are called moose) than it does in Europe. Additionally, elk in North America is *Cervus elaphus canadensis* so it can be extra confusing for red deer in Europe.

Additionally, SI Table 5 and SI 6 were unavailable. I was only able to see SI 1-4, please provide them.

Page 2 Line 56: "ZoomS is the method of choice..." I disagree that ZoomS is "the" method of choice for species ID. LC-MS provides species information and depth of characterization at the same time and can utilize proteins other than collagen I for species ID. The following sentence also refutes this one. An additional point, there is no such thing as "collagen," all collagens must be denoted by their type (e.g., collagen I).

Page 3 Line 22-24: Was the ring found in the brown gyttja layer? Was the layer it was found in reworked? If so, please clarify here.

Page 3 Line 52: Should pixel be voxel for this CT data? Usually each data point in CT data is a 3D point (i.e., a voxel) and not a 2D point (pixel).

Page 3 Line 55-56: "The resulting 3D volumes are cylinders with a diameter and height of 3.2 cm and 1.35 cm respectively corresponding to the different pixel resolution." This does not make sense to me. Please clarify how differences in voxel size result in different cylinders of the same object.

Page 3 Line 57-60: Why were these data not analyzed using a more standard approach to CT data? There are open access options (e.g., imageJ and boneJ) that can open voxel data and allow for analysis. ImageJ also allows for segmentation of the data and 3D visualization is possible as well. Please re-analyze the data with one of these tools.

Page 4 Line 17-20: "These samples were taken from a specimen that was defleshed by heating it in water for three days at c. 65°C. This defleshing process means that it has an equivalent thermal age (22) of around 2.5 kya at Syltholm, (assuming an activation energy of 173kJ mol⁻¹ for collagen degradation at the site whilst buried in sediment on the seafloor at a mean annual temperature of 7.5°C)," Please remove reference to thermal age. This sample is better described as an experimentally heated extant sample for comparison. Additionally, 65C is not going to drastically change this sample over 3 days, so I don't see how it could reflect 2.5 kya of aging.

Page 4 Line 26-36: Why were the first supernatants discarded? Schroeter et al. 2016 has shown clearly that discarding supernatants during bone protein extraction is ill advised because it leads to a loss of information.

Schroeter, E. R.; DeHart, C. J.; Schweitzer, M. H.; Thomas, P. M.; Kelleher, N. L., Bone protein extractomics: comparing the efficiency of bone protein extractions of *Gallus gallus* in tandem mass spectrometry, with an eye towards paleoproteomics. *PeerJ* 2016, 4, e2603.

Additionally, what were the protein concentrations after extraction? This concentration impacts the ratio of trypsin to protein and the resultant digestion completeness.

Page 5 Line 18-20: Please describe all of the mass spectrometry parameters that were used here as well (In short form is fine).

Page 5 Line 23-26: Why was *Bos taurus* used to fill in missing data on Cervid data instead of the more closely related and available white-tailed deer collagen sequences from the white-tailed deer genome?

Page 5 Line 42-49: String networks are not particularly meaningful here because basically all of the detected proteins are derived from the ECM. They also mis-split the protein groups because bone has many of the serum proteins as part of the ECM. I would recommend removing them and only including the Venn diagram.

Page 5 Line 59-60: Unless you have additional context for stable isotope species restriction, you can't say anything other than that this is a C3 browser. Please revise.

Page 6 Line 3-15: Is there any evidence of mixing in this layer? It was unclear in the stratigraphic description. If there is mixing then a safer restricted space is likely the full age range detected here. If there is more evidence to conclude they are similar age, please add it. If there is no evidence of mixing, then the as written age makes sense.

Page 6 Line 19-25: How can you tell what are primary and secondary osteons from these uCT data? These data do not have the resolution do this. Histological analysis would likely be better in this case (but may not be possible with sampling restrictions). Additionally, please add the normal diameter for deer bone vasculature (only antler was reported).

Page 6 Line 47-48: "Given the spectral similarity between the two extractions, it is likely that the artefact was not contaminated with humic acids from the environment" If humics were present, what differences would you expect to see? What color were the extractions? Is it possible the humic substances are generating ion suppression reducing the intensity of your Extraction 1 sample that you are recovering in Extraction 2?

Page 7 Line 5: Why is collagen I alpha 1 and alpha 2 combined? That is atypical for data searching. Also was the searched sequence only the mature collagen I form or does it contain the propeptides?

Page 7 Line 8-9: "blood proteins (alpha 2-HS glycoprotein (AHSG), pigment epithelium-derived factor/ serpin family F member 1 (SERPINF1), immunoglobulin gamma-1 heavy chain (IGHG1), serum albumin (ALB), apolipoprotein A-I (APOA1), and nucleobindin 1 (NUCB1))" AHSG is associated with mineral and PEDF is secreted by osteoblasts, so aren't really blood proteins in this case. Please revise.

Page 7 Line 54: "However, the plasma protein APOA1 was found to be uniquely shared between the ring and the heated modern red deer antler" APOA1 is not uncommon in other bone studies, so it is best to not assume it is specific for antler. (e.g., Schroeter, E. R.; DeHart, C. J.; Schweitzer, M. H.; Thomas, P. M.; Kelleher, N. L., Bone protein extractomics: comparing the efficiency of bone protein extractions of *Gallus gallus* in tandem mass spectrometry, with an eye towards paleoproteomics. *PeerJ* 2016, 4, e2603., Cleland, T. P., Human Bone Paleoproteomics Utilizing the Single-Pot, Solid-Phase-Enhanced Sample Preparation Method to Maximize Detected Proteins and Reduce Humics. *Journal of Proteome Research* 2018, 17, (11), 3976-3983., Sawafuji, R.; Cappellini, E.; Nagaoka, T.; Fotakis, A. K.; Jersie-Christensen, R. R.; Olsen, J. V.; Hirata, K.; Ueda, S., Proteomic profiling of archaeological human bone. *Royal Society Open Science* 2017, 4, (6), 161004.)

Page 8 Line 9-10: "This process occurs naturally over time, and while accelerated by factors such as temperature (38), these results indicate that the proteins examined are authentic due to their greater amount of damage compared to modern contamination" Deamidation is more complex

than just a temperature related change (see Schroeter, E. R.; Cleland, T. P., Glutamine deamidation: an indicator of antiquity, or preservational quality? *Rapid Communications in Mass Spectrometry* 2016, 30, 251-255.) please clarify this here.

Page 9 Line 7-10: "For red deer, up to 30 kg of antler can be generated within 3 months (45), increasing the probability of recovering bone formation and mineralisation proteins from this tissue, compared to mature long bone" Unless there is evidence for up-regulation in these proteins in extant studies, this sentence just isn't correct. All of these proteins are routinely detected in bone studies, so detection probability has nothing to do with antler/bone.

Page 9 Line 10-13: "In addition, THBS1 and COL1, along with fibrillin-1, are required for the permanence of matrix scaffold construction in the initial extracellular osseous matrix formation (46), therefore, the presence of THBS1 is suggestive of maturing bony matrix at an early stage of development." THBS1 is also associated with bone homeostasis (<https://asbmr.onlinelibrary.wiley.com/doi/abs/10.1002/jbmr.2308>) so does not strictly relate to early development as described.

Page 9 Line 16: If you haven't quantified how much Col11 and 12 is present then their presence here really does not support an antler origin because bone has both proteins as well.

Page 9 Line 21-22: "identification is also supported by the presence of SERPINF1, APOA1, POSTN, and THBS1, all of which have been implicated in axon/nerve growth in growing antlers" None of these proteins are unique to antler (including in the literature) and are all consistently detected from bone samples. PEDF is expressed by osteoblasts, APOA1 is from blood (see citations above), periostin is produced by osteoblasts, and thrombospondin 1 is important for bone homeostasis. Please revise.

Page 9 Line 27-29: "An archaeological study of human remains from different biological ages only discovered APOA1 in infants (59), indicating its preferential presence archaeologically in plasma rich, developing osseous tissue" Cleland et al. 2018 found APOA1 in adult human individuals so age does not really have much to do with APOA1 presence or absence (Cleland, T. P., Human Bone Paleoproteomics Utilizing the Single-Pot, Solid-Phase-Enhanced Sample Preparation Method to Maximize Detected Proteins and Reduce Humics. *Journal of Proteome Research* 2018, 17, (11), 3976-3983.)

Page 9 Line 33-36: "the genes of three of the top five proteins with the most peptides recovered, collagen types 1 and 3 and osteocalcin, have been shown to be expressed in the ossified part of velvet antler 10-30 times more than in the skeleton of the animal" To support quantitative statements from the literature, quantification of protein amounts are necessary here as well. Qualitative detection of typically abundant proteins in bone \neq quantitative assessments of how much is present.

Page 9 Line 39: High sequence coverage of collagen I from limited starting bone material is not surprising and is fairly typical for recent LC-MS studies. Please revise.

Page 9 Line 46: "heating in water - a commonly used surrogate for diagenesis (66)" Cooked bone is very different from 65C heating for 3 days. Please do not make this comparison.

Page 9 Line 48: "As expected the use of thermally degraded samples yielded proteomes of similar size to that of the ring" The proteome sizes detected here are likely more an expression of the extractome of this extraction method than anything about diagenesis. Please revise.

Page 10 Line 3: "tentative evidence that the Syltholm ring was manufactured from red deer antler" I'm not at all convinced that this ring came from antler and not bone. Using only qualitative data, none of the proteins detected from the ring can directly distinguish between the two bone tissues. Please revise accordingly.

Page 10 Lines 27-31: Again, because I'm not convinced that this ring is derived from antler, hypothesizing manufacture is premature. Please revise.

Decision letter (RSOS-191172.R0)

28-Aug-2019

Dear Mr Jensen,

The editors assigned to your paper ("The biomolecular characterisation of an Early Neolithic finger ring from Syltholm, Denmark") have now received comments from reviewers. We would like you to revise your paper in accordance with the referee and Associate Editor suggestions which can be found below (not including confidential reports to the Editor). Please note this decision does not guarantee eventual acceptance.

Please submit a copy of your revised paper before 20-Sep-2019. Please note that the revision deadline will expire at 00.00am on this date. If we do not hear from you within this time then it will be assumed that the paper has been withdrawn. In exceptional circumstances, extensions may be possible if agreed with the Editorial Office in advance. We do not allow multiple rounds of revision so we urge you to make every effort to fully address all of the comments at this stage. If deemed necessary by the Editors, your manuscript will be sent back to one or more of the original reviewers for assessment. If the original reviewers are not available, we may invite new reviewers.

- Data accessibility

If you wish to submit your supporting data or code to Dryad (<http://datadryad.org/>), or modify your current submission to dryad, please use the following link:
<http://datadryad.org/submit?journalID=RSOS&manu=RSOS-191172>

- Competing interests

- Authors' contributions

- Acknowledgements

- Funding statement

Kind regards,

Andrew Dunn

on behalf of Professor Diwakar Shukla (Associate Editor) and Kevin Padian (Subject Editor)
 openscience@royalsociety.org

Editor Comments:

Thanks for your submission. As you will see the reviewers had a great many comments that need to be addressed. Please respond to each of them in your revision. If you find you need more time, ask our editorial office and it will not be a problem.

Comments to Author:

Reviewers' Comments to Author:

Reviewer: 1

Comments to the Author(s)

The manuscript is excellent but wordy; try to avoid repetition and redundant information. The following specific suggestions may improve the paper.

Abstract:

if you are confident that your results identify the raw material as antler rather than bone (e.g. p.10 line 59), I would mention this in the abstract.

Use-wear (e.g. p.10 line 33):

why would you expect to see use-wear on the internal surface of the ring, even if it was used? Is the ring in a more pristine condition than those discussed in para.3 of the introduction? If the ring was in pristine condition, indicating that it might have broken during manufacture (p.2 line 20), why not mention absence of use-wear on the external (polished) surface?

Micro-CT:

I was unable to open the supplementary video – perhaps the format is too restrictive. The results appear to exclude ivory as well as enamel or dentine – this should probably be mentioned in the discussion (e.g. p.8 line 37).

Dating:

I am not convinced that the broken ring fragment is necessarily the same date as the closest wooden artefact dated (p.2 line 26, p.6 line 12, Fig. 2). No evidence has been presented except proximity, but the taphonomic situation and the fact that both finds are broken and incomplete means that there's a high risk that their proximity is coincidental. The ring, like the harpoon fragment in the same deposit, might even be Mesolithic.

A better way to estimate the date of any undated find from this deposit is to use the scatter of dates on all artefactual material to derive a single date range. The easiest method is to add one line of code to the OxCal model, within the bounded phase:

Date ("ring");

This function (see Bronk Ramsey 2009, your ref.19) generates a probability density function for the date of an event between the onset and termination boundaries you have calculated based on the spread of dates from wooden artefacts. It would still be misleading if the ring was re-deposited from older layers, but if it was freshly deposited in the gyttja during the period that wooden spears etc were discarded at this location, it would be a more realistic estimate than just picking one spear's date.

The dating results section is contradictory: "The harpoon was excluded from the model due [to] a low statistical agreement in the Bayesian model (see 17). The reason for this is unknown as all

collagen quality parameters are within expected ranges." The reason is obvious; you got the right date on the harpoon (given the Late Mesolithic typological dating mentioned earlier in the paragraph) but the wooden artefacts belong to a later (early Neolithic) phase of occupation. There are plenty of ways in which a fragment of an older harpoon could have been deposited at this location during the early Neolithic. The problem is therefore why you should be confident that the ring fragment was freshly deposited, and that it belongs to the same phase of activity as the wooden artefacts.

Also, "The dates demonstrate that the site was frequented both before and after the Mesolithic/Neolithic transition" is true (assuming a transition at c.4000 cal BC, which should have been indicated in para.1 of the introduction), but "The usage period of the site was estimated using a simple Bayesian model ... The onset of activity at the site is estimated to be c. 6060 cal. BP" is misleading, in the sense that this model applies only to the 15 wooden artefacts at this specific location, and not to Syltholm in general. Simply relying on these 15 dates I would not be confident that there was a Mesolithic phase at Syltholm at all.

LC-MS/MS methods

Some clarification may be required here. The fact that "At present, no proteins for elk have been uploaded to UniProt" seems to imply that you could not actually exclude elk, yet in the results (p.6 line 54) you talk about 3 known SAPs which can be used to separate deer from elk. Where did the elk reference data come from to support this claim? Incidentally, it is impossible for a non-specialist to see how Fig 5 shows the results indicated in the text.

Figure 6 caption "Venn diagram demonstrating 5 proteins shared between the ring and antler, and 0 proteins shared between the ring and bone", but figure shows 10 proteins shared by all 3 samples, and only 1 shared between the ring and antler (but not the bone).

LC-MS/MS protein authentication

What are contaminant proteins, and how do you know that they are contaminants? I can see that from the low deamidation rate (Figure 7) they should not be from the ring itself, but where does this contamination come from?

Minor text edits:

p.2 line 8: do not use "bp" for dates (in the 1980s it was used in archaeological literature for uncalibrated radiocarbon ages). Here you mean "cal BP" (as applied in the rest of the paper), which would be correct. However, Neolithic archaeology in Denmark tends to use the cal BC scale, and unless you feel strongly that this practice should change, I would convert all the calibrated dates into cal BC, including in Fig. 2.

p.2 line 10: "exotic objects, including a T-shaped antler axe" – T-axes are quintessentially Ertebølle artefacts, so I don't see why you regard this one as exotic

p.2 lines 11-12: "... Arkadenrand and Stichband type ceramics. These finds suggest connections with Neolithic societies of northern Germany" – perhaps superfluous information here, but the former would date to the mid-4th millennium cal BC while the latter is much older (early 5th millennium cal BC). Since there is no contextual association between these imports and the osseous ring, I am not sure it's relevant, but perhaps insert "of the 4th and 5th millennia cal BC" after "northern Germany". The confusing aspect for readers is that at Syltholm the 5th millennium is still the Mesolithic, as alluded to in the next paragraph. I think it would be clearer if you just delete the text "This indicates that .. and the beginning of the Funnel Beaker Culture" (p.2 lines 23-24) and insert Mesolithic and Neolithic in the first sentence of para.1, i.e. "...belonging to the Mesolithic Ertebølle... and Early Neolithic Funnel Beaker periods".

The third paragraph of the introduction gives many potential parallels for the ring elsewhere in Europe, but no absolute dates. Non-European archaeologists (and other readers) would probably not realise that Anatolian Neolithic could be much earlier, that LBK and Rubané are the same thing and date to the late 6th millennium cal BC, that Middle Neolithic in the examples given is probably 3rd millennium, etc.

p.3 lines 10-16: I found this description confusing (assuming that I was unfamiliar with the site and the project, and could not read your ref.15, which is in Danish); much of this information is probably redundant for the purpose of this paper.

p.4 line 28: "They were then vortexed" – laboratory colloquialism!

p.5 line 26: "all proteins available for red deer were downloaded" – another colloquialism; data were downloaded or uploaded, not proteins.

"searched": there are several instances (e.g. in "The Thermo RAW files generated were then searched using the software MaxQuant") in which this term (or similar expressions such as "peak picked") are used colloquially or ungrammatically. I think you mean that you used software X to look for reference spectra closely matching the data files.

p.5 line 48: "Groupings were generated by applying kmeans clustering (number of clusters = 3)" – Hyphenate k-means. Groupings of what? Why 3 clusters?

p.8, line 45ff: These sentences "However, the most likely candidate, based on the ZooMS data, would be red deer since elk was part of the depauperate fauna (39). Red deer were still abundant in Denmark during the Atlantic period, as opposed to elk, which disappeared following the Littorina transgressions that turned Denmark into an archipelago." should be re-written, because (1) the argument is not based on the ZooMS results, but on biogeography; (2) "depauperate" is not a common word ("impoverished" would be more widely understood) and the implicit point is that elk was not part of the impoverished Danish fauna; (3) the Littorina transgression is not dated; (4) the Atlantic period is not dated (I know what you mean, but many readers will not, and might assume that the Atlantic period was initiated by the Littorina transgression). A much simpler chain of reasoning might be that there is abundant red deer bone at Syltholm but no elk bones have been identified, if you know this to be the case.

Table 1 and Figure 2: what is a "fishing ruse"?

Reviewer: 2

Comments to the Author(s)

In this manuscript, Jensen et al. attempt to determine the species and skeletal element of origin of an osseous ring from Denmark. They perform μ CT, ZooMS, and LC-MS to evaluate it. This is an interesting study, but has some major interpretation issues that make its conclusions unconvincing. Aspects of the experimental setup also limit the conclusions as well.

One minor thing to revise throughout, for clarity throughout please refer to red deer and elk by their scientific names only. Elk has a very different meaning in North America (where they are called moose) than it does in Europe. Additionally, elk in North America is *Cervus elaphus canadensis* so it can be extra confusing for red deer in Europe.

Additionally, SI Table 5 and SI 6 were unavailable. I was only able to see SI 1-4, please provide them.

Page 2 Line 56: "ZoomMS is the method of choice..." I disagree that ZoomMS is "the" method of choice for species ID. LC-MS provides species information and depth of characterization at the same time and can utilize proteins other than collagen I for species ID. The following sentence also refutes this one. An additional point, there is no such thing as "collagen," all collagens must be denoted by their type (e.g., collagen I).

Page 3 Line 22-24: Was the ring found in the brown gyttja layer? Was the layer it was found in reworked? If so, please clarify here.

Page 3 Line 52: Should pixel be voxel for this CT data? Usually each data point in CT data is a 3D point (i.e., a voxel) and not a 2D point (pixel).

Page 3 Line 55-56: "The resulting 3D volumes are cylinders with a diameter and height of 3.2 cm and 1.35 cm respectively corresponding to the different pixel resolution." This does not make sense to me. Please clarify how differences in voxel size result in different cylinders of the same object.

Page 3 Line 57-60: Why were these data not analyzed using a more standard approach to CT data? There are open access options (e.g., imageJ and boneJ) that can open voxel data and allow for analysis. ImageJ also allows for segmentation of the data and 3D visualization is possible as well. Please re-analyze the data with one of these tools.

Page 4 Line 17-20: "These samples were taken from a specimen that was defleshed by heating it in water for three days at c. 65°C. This defleshing process means that it has an equivalent thermal age (22) of around 2.5 kya at Syltholm, (assuming an activation energy of 173kJ mol⁻¹ for collagen degradation at the site whilst buried in sediment on the seafloor at a mean annual temperature of 7.5°C)," Please remove reference to thermal age. This sample is better described as an experimentally heated extant sample for comparison. Additionally, 65C is not going to drastically change this sample over 3 days, so I don't see how it could reflect 2.5 kya of aging.

Page 4 Line 26-36: Why were the first supernatants discarded? Schroeter et al. 2016 has shown clearly that discarding supernatants during bone protein extraction is ill advised because it leads to a loss of information.

Schroeter, E. R.; DeHart, C. J.; Schweitzer, M. H.; Thomas, P. M.; Kelleher, N. L., Bone protein extractomics: comparing the efficiency of bone protein extractions of *Gallus gallus* in tandem mass spectrometry, with an eye towards paleoproteomics. *PeerJ* 2016, 4, e2603.

Additionally, what were the protein concentrations after extraction? This concentration impacts the ratio of trypsin to protein and the resultant digestion completeness.

Page 5 Line 18-20: Please describe all of the mass spectrometry parameters that were used here as well (In short form is fine).

Page 5 Line 23-26: Why was *Bos taurus* used to fill in missing data on Cervid data instead of the more closely related and available white-tailed deer collagen sequences from the white-tailed deer genome?

Page 5 Line 42-49: String networks are not particularly meaningful here because basically all of the detected proteins are derived from the ECM. They also mis-split the protein groups because bone has many of the serum proteins as part of the ECM. I would recommend removing them and only including the Venn diagram.

Page 5 Line 59-60: Unless you have additional context for stable isotope species restriction, you can't say anything other than that this is a C3 browser. Please revise.

Page 6 Line 3-15: Is there any evidence of mixing in this layer? It was unclear in the stratigraphic description. If there is mixing then a safer restricted space is likely the full age range detected here. If there is more evidence to conclude they are similar age, please add it. If there is no evidence of mixing, then the as written age makes sense.

Page 6 Line 19-25: How can you tell what are primary and secondary osteons from these uCT data? These data do not have the resolution do this. Histological analysis would likely be better in this case (but may not be possible with sampling restrictions). Additionally, please add the normal diameter for deer bone vasculature (only antler was reported).

Page 6 Line 47-48: "Given the spectral similarity between the two extractions, it is likely that the artefact was not contaminated with humic acids from the environment" If humics were present, what differences would you expect to see? What color were the extractions? Is it possible the humic substances are generating ion suppression reducing the intensity of your Extraction 1 sample that you are recovering in Extraction 2?

Page 7 Line 5: Why is collagen I alpha 1 and alpha 2 combined? That is atypical for data searching. Also was the searched sequence only the mature collagen I form or does it contain the propeptides?

Page 7 Line 8-9: "blood proteins (alpha 2-HS glycoprotein (AHSG), pigment epithelium-derived factor/ serpin family F member 1 (SERPINF1), immunoglobulin gamma-1 heavy chain (IGHG1), serum albumin (ALB), apolipoprotein A-I (APOA1), and nucleobindin 1 (NUCB1))" AHSG is associated with mineral and PEDF is secreted by osteoblasts, so aren't really blood proteins in this case. Please revise.

Page 7 Line 54: "However, the plasma protein APOA1 was found to be uniquely shared between the ring and the heated modern red deer antler" APOA1 is not uncommon in other bone studies, so it is best to not assume it is specific for antler. (e.g., Schroeter, E. R.; DeHart, C. J.; Schweitzer, M. H.; Thomas, P. M.; Kelleher, N. L., Bone protein extractomics: comparing the efficiency of bone protein extractions of Gallus gallus in tandem mass spectrometry, with an eye towards paleoproteomics. PeerJ 2016, 4, e2603., Cleland, T. P., Human Bone Paleoproteomics Utilizing the Single-Pot, Solid-Phase-Enhanced Sample Preparation Method to Maximize Detected Proteins and Reduce Humics. Journal of Proteome Research 2018, 17, (11), 3976-3983., Sawafuji, R.; Cappellini, E.; Nagaoka, T.; Fotakis, A. K.; Jersie-Christensen, R. R.; Olsen, J. V.; Hirata, K.; Ueda, S., Proteomic profiling of archaeological human bone. Royal Society Open Science 2017, 4, (6), 161004.)

Page 8 Line 9-10: "This process occurs naturally over time, and while accelerated by factors such as temperature (38), these results indicate that the proteins examined are authentic due to their greater amount of damage compared to modern contamination" Deamidation is more complex than just a temperature related change (see Schroeter, E. R.; Cleland, T. P., Glutamine deamidation: an indicator of antiquity, or preservational quality? Rapid Communications in Mass Spectrometry 2016, 30, 251-255.) please clarify this here.

Page 9 Line 7-10: "For red deer, up to 30 kg of antler can be generated within 3 months (45), increasing the probability of recovering bone formation and mineralisation proteins from this tissue, compared to mature long bone" Unless there is evidence for up-regulation in these

proteins in extant studies, this sentence just isn't correct. All of these proteins are routinely detected in bone studies, so detection probability has nothing to do with antler/bone.

Page 9 Line 10-13: "In addition, THBS1 and COL1, along with fibrillin-1, are required for the permanence of matrix scaffold construction in the initial extracellular osseous matrix formation (46), therefore, the presence of THBS1 is suggestive of maturing bony matrix at an early stage of development." THBS1 is also associated with bone homeostasis (<https://asbmr.onlinelibrary.wiley.com/doi/abs/10.1002/jbmr.2308>) so does not strictly relate to early development as described.

Page 9 Line 16: If you haven't quantified how much Col11 and 12 is present then their presence here really does not support an antler origin because bone has both proteins as well.

Page 9 Line 21-22: "identification is also supported by the presence of SERPINF1, APOA1, POSTN, and THBS1, all of which have been implicated in axon/nerve growth in growing antlers" None of these proteins are unique to antler (including in the literature) and are all consistently detected from bone samples. PEDF is expressed by osteoblasts, APOA1 is from blood (see citations above), periostin is produced by osteoblasts, and thrombospondin 1 is important for bone homeostasis. Please revise.

Page 9 Line 27-29: "An archaeological study of human remains from different biological ages only discovered APOA1 in infants (59), indicating its preferential presence archaeologically in plasma rich, developing osseous tissue" Cleland et al. 2018 found APOA1 in adult human individuals so age does not really have much to do with APOA1 presence or absence (Cleland, T. P., Human Bone Paleoproteomics Utilizing the Single-Pot, Solid-Phase-Enhanced Sample Preparation Method to Maximize Detected Proteins and Reduce Humics. *Journal of Proteome Research* 2018, 17, (11), 3976-3983.)

Page 9 Line 33-36: "the genes of three of the top five proteins with the most peptides recovered, collagen types 1 and 3 and osteocalcin, have been shown to be expressed in the ossified part of velvet antler 10-30 times more than in the skeleton of the animal" To support quantitative statements from the literature, quantification of protein amounts are necessary here as well. Qualitative detection of typically abundant proteins in bone \neq quantitative assessments of how much is present.

Page 9 Line 39: High sequence coverage of collagen I from limited starting bone material is not surprising and is fairly typical for recent LC-MS studies. Please revise.

Page 9 Line 46: "heating in water - a commonly used surrogate for diagenesis (66)" Cooked bone is very different from 65C heating for 3 days. Please do not make this comparison.

Page 9 Line 48: "As expected the use of thermally degraded samples yielded proteomes of similar size to that of the ring" The proteome sizes detected here are likely more an expression of the extractome of this extraction method than anything about diagenesis. Please revise.

Page 10 Line 3: "tentative evidence that the Syltholm ring was manufactured from red deer antler" I'm not at all convinced that this ring came from antler and not bone. Using only qualitative data, none of the proteins detected from the ring can directly distinguish between the two bone tissues. Please revise accordingly.

Page 10 Lines 27-31: Again, because I'm not convinced that this ring is derived from antler, hypothesizing manufacture is premature. Please revise.

Author's Response to Decision Letter for (RSOS-191172.R0)

See Appendix A.

RSOS-191172.R1 (Revision)

Review form: Reviewer 2

Is the manuscript scientifically sound in its present form?

Yes

Are the interpretations and conclusions justified by the results?

Yes

Is the language acceptable?

Yes

Do you have any ethical concerns with this paper?

No

Have you any concerns about statistical analyses in this paper?

No

Recommendation?

Accept as is

Comments to the Author(s)

I thank the authors for their thoughtful responses. This will be an exciting contribution to the field.

Decision letter (RSOS-191172.R1)

22-Nov-2019

Dear Mr Jensen,

It is a pleasure to accept your manuscript entitled "The biomolecular characterisation of a finger ring contextually dated to the emergence of the Early Neolithic from Syltholm, Denmark" in its current form for publication in Royal Society Open Science. The comments of the reviewer(s) who reviewed your manuscript are included at the foot of this letter.

Please ensure that you send to the editorial office an editable version of your accepted manuscript, and individual files for each figure and table included in your manuscript. You can

send these in a zip folder if more convenient. Failure to provide these files may delay the processing of your proof. You may disregard this request if you have already provided these files to the editorial office.

Kind regards,

on behalf of Professor Diwakar Shukla (Associate Editor) and Kevin Padian (Subject Editor)
openscience@royalsociety.org

Reviewer comments to Author:

Reviewer: 2
Comments to the Author(s)

I thank the authors for their thoughtful responses. This will be an exciting contribution to the field.

Appendix A

Authors' response to the reviewer comments

We would like to thank the reviewers for the many useful comments and corrections. We have strived to address them all and now feel the manuscript is much stronger than the previous version.

To aid in your assessment of the corrections we have highlighted all changes in yellow in this document as well as uploading a separate version of the manuscript. Also we have demarked our responses (red) from the reviewers comments (black).

Reviewer: 1

Comments to the Author(s)

The manuscript is excellent but wordy; try to avoid repetition and redundant information. The following specific suggestions may improve the paper.

Abstract:

Reviewer comment: if you are confident that your results identify the raw material as antler rather than bone (e.g. p.10 line 59), I would mention this in the abstract.

Authors response: Unfortunately, due to the limited knowledge of bone vs antler proteomes, we cannot be certain of this assignment, and only assert that antler is consistent with the data we have at present, but bone cannot be ruled out. Hence why we do not state it in the abstract as fact. This has been stated more clearly in the discussion (pg 9-10).

Reviewer comment: Use-wear (e.g. p.10 line 33):

why would you expect to see use-wear on the internal surface of the ring, even if it was used?

Authors' response: As mentioned in the introduction, the interior of the ring still shows well-preserved traces of carving. Prolonged wearing of the ring would, in effect, polish and wear down the internal scrape marks left by the carving of the ring, creating a smooth appearance.

Reviewer comment: Is the ring in a more pristine condition than those discussed in para.3 of the introduction? If the ring was in pristine condition, indicating that it might have broken during manufacture (p.2 line 20), why not mention absence of use-wear on the external (polished) surface?

Action taken: In order to rectify this apparent contradiction, and emphasize the absence of use-wear on the external surface, we have slightly reworded a sentence in the introduction: "The exterior is finely polished, with only microscopic scratches and no use-wear visible, while the interior still shows well-preserved traces of carving, suggesting that it was either

barely worn, or that it broke during manufacture.” and the sentence at the bottom of page 10 to read “Apart from being broken, the ring was well preserved and does not show any evidence of use-wear, which suggests that it was produced in the vicinity of the site.”

Micro-CT:

Reviewer comment: I was unable to open the supplementary video – perhaps the format is too restrictive. The results appear to exclude ivory as well as enamel or dentine – this should probably be mentioned in the discussion (e.g. p.8 line 37).

Action taken: We thank the author for this observation. Ivory is mainly composed of dentine, therefore this implies that ivory is also not a material candidate. We have made this more clear by adding the following on page 6: “The diameter of osteons in antler ranges from 100–225 µm diameter (33), while enamel only contains nano-pores, and tubules in dentine and ivory (mostly composed of dentine) are approximately 1-2 µm in diameter (34).” and also mention it on page 8 where the reviewer indicates: “Enamel and bone/antler are easily distinguished from one another by non-destructive Micro CT through visualisation of Haversian canals, which are absent in enamel and dentine (including ivory)”.

As for the file format of the supplementary video, we apologise for this inconvenience, we have now supplied it as a more accessible format (AVI).

Dating:

Reviewer comment: I am not convinced that the broken ring fragment is necessarily the same date as the closest wooden artefact dated (p.2 line 26, p.6 line 12, Fig. 2). No evidence has been presented except proximity, but the taphonomic situation and the fact that both finds are broken and incomplete means that there’s a high risk that their proximity is coincidental. The ring, like the harpoon fragment in the same deposit, might even be Mesolithic.

Authors’ response: We acknowledge that there is a chance the ring is Late Mesolithic based on fact, that a limited amount of 14C dates derive from this period. To strengthen our argument for it being Neolithic, we included an additional 55 dates from the entire site. We then created a KDE model based on the total 70 to infer periods of activity. The vast majority of 14C dates are Early Neolithic. In addition, only two pieces of Ertebølle ceramics has been found on this specific site, while several complete vessels and hundreds of Neolithic sherds has been found.

Action taken: text in method section radiocarbon dating (pg3) changed to “However, 70 radiocarbon dates of various artefacts from the site (Table 1), were commissioned as part of the wider project carried out by the Museum Lolland-Falster.”

Figure 2 updated to include KDE. Caption updated: “**Fig. 2.** Probability distributions of the 15 radiocarbon samples found in close proximity to the ring. The colored probability distributions are the result of a simple Bayesian model assuming all sample to originate from

the same phase of activity. Onset and termination of the phase are indicated with black probability distributions. The light green probability distribution is the date we propose for the ring as well, based on the proximity of the spear to the ring. KDE model of all 70 ¹⁴C dates indicating a single inferred period of archaeological activity at site in general.”

In the results section *Radiocarbon dates* (pg6) the text has been updated: “Due to their close contextual and stratigraphic association, we propose that the ring has a similar date, but we cannot rule out that the ring is Late Mesolithic. However, as shown in the kernel density estimation (KDE) model, which includes an additional 55 dates from the surrounding area, activity at the site in general peaks during the Early Neolithic (Fig. 2).”

In the discussion section *The Syltholm finger ring is made from Cervus antler or bone* (pg 8) the text has been updated: “Bayesian modelling of 15 radiocarbon dates, obtained from the immediate proximity to the ring was used to indirectly date the ring to the Early Neolithic. Additionally, KDE modelling utilising a total of 70 dates confirmed that whilst the site was frequented in the Mesolithic the most intense activity was in the Early Neolithic.”

Reviewer comment: A better way to estimate the date of any undated find from this deposit is to use the scatter of dates on all artefactual material to derive a single date range. The easiest method is to add one line of code to the OxCal model, within the bounded phase: Date ("ring");

This function (see Bronk Ramsey 2009, your ref.19) generates a probability density function for the date of an event between the onset and termination boundaries you have calculated based on the spread of dates from wooden artefacts. It would still be misleading if the ring was re-deposited from older layers, but if it was freshly deposited in the gyttja during the period that wooden spears etc were discarded at this location, it would be a more realistic estimate than just picking one spear's date.

Action taken: The text has now been changed to reflect that the broken ring is found within the modelled phase and hence that the best age estimate we can provide is that it dates in the range between the phase onset and the phase termination. We have changed to text on page 8 to:

“The reason for this is unknown as all collagen quality parameters are within expected ranges, however, it is possible that harpoon has been redeposited. The ring dates to the usage period of the site (c. 6060 calBP - c. 5590 calBP), however, the ash (*Fraxinus excelsior*) spear, found closest to the ring returned a date of 5,983-5,750 cal. BP (5,128±35 ¹⁴C yr BP). Due to their close contextual and stratigraphic association, we propose that the ring has a similar date, but we can not rule out that the ring is Late Mesolithic. However, as shown in the kernel density estimation (KDE) model, which includes an additional 55 dates from the surrounding area, activity at the site in general peaks during the Early Neolithic (Fig. 2).”

Reviewer comment: The dating results section is contradictory: “The harpoon was excluded from the model due [to] a low statistical agreement in the Bayesian model (see 17). The reason for this is unknown as all collagen quality parameters are within expected ranges.” The reason is obvious; you got the right date on the harpoon (given the Late Mesolithic typological dating mentioned earlier in the paragraph) but the wooden artefacts belong to a later (early Neolithic) phase of occupation. There are plenty of ways in which a fragment of an older harpoon could have been deposited at this location during the early Neolithic. The problem is therefore why you should be confident that the ring fragment was freshly deposited, and that it belongs to the same phase of activity as the wooden artefacts.

Action taken: The text on page 6 has been changed to: “The harpoon was excluded from the model due to a low statistical agreement in the Bayesian model (see 17). The reason for this is unknown as all collagen quality parameters are within expected ranges, however, it is possible that harpoon has been redeposited.”

Reviewer comment: Also, “The dates demonstrate that the site was frequented both before and after the Mesolithic/Neolithic transition” is true (assuming a transition at c.4000 cal BC, which should have been indicated in para.1 of the introduction), but “The usage period of the site was estimated using a simple Bayesian model ... The onset of activity at the site is estimated to be c. 6060 cal. BP” is misleading, in the sense that this model applies only to the 15 wooden artefacts at this specific location, and not to Syltholm in general. Simply relying on these 15 dates I would not be confident that there was a Mesolithic phase at Syltholm at all.

Action taken: We have added to figure 2 a panel showing the KDE modelled activity from the location. This model is based on 70 14C samples found during the excavation of Syltholm 906-II. Only very few and scattered finds are dating outside the early Neolithic period, however, the main feature which stands out is that the activity was very intense during the early Neolithic and highly sporadic in other periods. We have also changed the caption slightly, and marked the changes in yellow.

In addition we have added all AMS dates to a separate SI file, now SI 2.

Fig. 2. Probability distribution of 15 radiocarbon samples found in close proximity to the ring. The colored probability distributions are the result of a simple Bayesian model assuming all sample to originate from the same phase of activity. Onset and termination of the phase are indicated with black probability distributions. The light green probability distribution is the date we propose for the ring as well, based on the proximity of the spear to the ring. KDE model of all 70 14C dates indicating periods activity.

In the discussion (pg 8) a sentence has been added “Additionally, KDE modelling utilising a total of 70 dates confirmed that whilst the site was frequented in the Mesolithic the most intense activity was in the Early Neolithic”

Additionally we have updated the title to “The biomolecular characterisation of a finger ring contextually dated to the emergence of the Early Neolithic from Syltholm, Denmark”

LC-MS/MS methods

Reviewer comment: Some clarification may be required here. The fact that “At present, no proteins for elk have been uploaded to UniProt” seems to imply that you could not actually exclude elk, yet in the results (p.6 line 54) you talk about 3 known SAPs which can be used to separate deer from elk. Where did the elk reference data come from to support this claim?

Action taken: The species identification comes from “previously published type 1 collagen sequences from red deer and elk (27)” (pg 5). The collagen peptides recovered are specific to the red deer, not elk, sequence in some key parts of the sequence. However, we do not have any other protein sequences for elk available to compare with this species identification from other proteins, and therefore, for the non-collagen type 1 proteins, we just indicate how specific they are based on current publically available data on UniProt. We have amended this to say (pg 5) “Furthermore, in order to identify proteins in addition to collagen type 1, all protein sequences available for *Cervus elaphus* were downloaded from UniProt (20-07-18). Unfortunately, no other proteins for *Alces* have, thus far, been uploaded to UniProt” and also emphasised this again later (pg 7): “In some cases, these proteins were identified specifically to *Cervus* (AHSG, IGHG1) or *Cervus/Odocoileus virginianus* (white-tailed deer) (SERPINF1, APOA1) when compared to all publicly available sequences for these proteins (which do not include sequences specific to *Alces*). Since white-tailed deer are native to the Americas, they can be excluded from species identification in this context. However, since the corresponding *Alces* sequences are not available, it cannot be ruled out that these may also match this species, and therefore results reflect current data availability. For the other identified proteins, we recovered peptides that were less species-specific (SI 3).”

Reviewer comment: Incidentally, it is impossible for a non-specialist to see how Fig 5 shows the results indicated in the text.

Action taken: After reviewing this part, we found an error regarding the positions of the SAPs. These have now been corrected.

The sites are position 741 P (*Cervus elaphus*) or A (*Alces alces*) on the $\alpha 1$ chain, and 454 P (*Cervus elaphus*) or I/L (*Alces alces*), and 749 S (*Cervus elaphus*) or T (*Alces alces*) on the $\alpha 2$ chain.

We have also updated the figure to include the amino acid position so this links with the text i.e. “The sites are position 758 P (*Cervus elaphus*) or A (*Alces alces*) on the $\alpha 1$ chain,”

Reviewer comment: Figure 6 caption “Venn diagram demonstrating 5 proteins shared between the ring and antler, and 0 proteins shared between the ring and bone”, but figure shows 10 proteins shared by all 3 samples, and only 1 shared between the ring and antler (but not the bone).

Action taken: We thank the reviewer for noting our mistake between two different versions of our analysis. This has been updated. In the main text we do not make this mistake.

Fig. 6. Venn-Diagrams: Venn diagram demonstrating 1 protein shared between the ring and antler, and 0 proteins shared between the ring and bone.

LC-MS/MS protein authentication

Reviewer comment: What are contaminant proteins, and how do you know that they are contaminants?

I can see that from the low deamidation rate (Figure 7) they should not be from the ring itself, but where does this contamination come from?

Action taken: Common contaminants are due to handling, storage and laboratory analysis. These are typically skin keratins and protinacious laboratory reagents e.g. bovine serum albumin. The authentication rational via demaidation is described in the text as well as being previously published. To make this more clear for those unfamiliar with palaeoproteomics, we have amended the text to include the phrase (pg 5) "In addition, the samples were searched against the MaxQuant contaminant database that identifies proteins that may be present due to sample handling and laboratory analysis. Any protein not considered authentic to the ring (i.e. keratins from skin, bovine serum albumin (used as a laboratory standard)) was not included in further analysis except as a comparison for deamidation levels."

Minor text edits:

Reviewer comment: p.2 line 8: do not use "bp" for dates (in the 1980s it was used in archaeological literature for uncalibrated radiocarbon ages). Here you mean "cal BP" (as applied in the rest of the paper), which would be correct. However, Neolithic archaeology in Denmark tends to use the cal BC scale, and unless you feel strongly that this practice should change, I would convert all the calibrated dates into cal BC, including in Fig. 2.

Action taken: We have now changed the text to; Ongoing excavations at Syltholm near Rødbyhavn on the island of Lolland, Denmark, have revealed an exceptionally well-preserved archaeological assemblage belonging to the Ertebølle (c. 7,350 - 5,950 cal BP) and Early Funnel Beaker periods (c. 5,950 - 4,750 cal BP).

Authors' response: It is correct that older literature tend to prefer cal BC, however, we believe that this is a matter of opinion. All fields within natural sciences use cal BP, while archaeology is in some instances the only discipline still using cal BC.

Reviewer comment: p.2 line 10: "exotic objects, including a T-shaped antler axe" – T-axes are quintessentially Ertebolle artefacts, so I don't see why you regard this one as exotic

Authors' response: T-shaped antler axes are indeed part of the Ertebølle inventory, but only in Western Denmark (Jutland and Funen) and Northern Germany. The "T" implies that the shaft-hole is placed in the middle of the antler, where as, in the western Ertebølle inventory (Zealand, Falster, Lolland and Møn) (termed *Rosenøkser*), the shaft hole is not placed in the middle but to the far end opposite the cutting edge. Before the excavations at Syltholm, only two other *true* T-antler axes had been found in eastern Denmark. Therefore describing this artefact as 'exotic' rather than 'rare' denotes that it originated from elsewhere.

Reviewer comment: p.2 lines 11-12: "... Arkadenrand and Stichband type ceramics. These finds suggest connections with Neolithic societies of northern Germany" – perhaps superfluous information here, but the former would date to the mid-4th millennium cal BC while the latter is much older (early 5th millennium cal BC). Since there is no contextual association between these imports and the osseous ring, I am not sure it's relevant, but perhaps insert "of the 4th and 5th millennia cal BC" after "northern Germany". The confusing aspect for readers is that at Syltholm the 5th millennium is still the Mesolithic, as alluded to in the next paragraph. I think it would be clearer if you just delete the text "This indicates that .. and the beginning of the Funnel Beaker Culture." (p.2 lines 23-24) and insert Mesolithic and Neolithic in the first sentence of para.1, i.e. "...belonging to the Mesolithic Ertebølle... and Early Neolithic Funnel Beaker periods".

Action taken: We have omitted the Stichband type. Further analyses by petro-mineralogy and geochemistry have yielded inconclusive results.

We have changed the text on page 2 to; The ring was found in a layer containing a large amount of wooden artefacts, which have been directly dated to between c. 6,300 - 5,500 cal BP (Table 1, Fig. 2), which mirrors the period of activity.

Reviewer comment: The third paragraph of the introduction gives many potential parallels for the ring elsewhere in Europe, but no absolute dates. Non-European archaeologists (and other readers) would probably not realise that Anatolian Neolithic could be much earlier, that LBK and Rubané are the same thing and date to the late 6th millennium cal BC, that Middle Neolithic in the examples given is probably 3rd millennium, etc.

Authors' response: It is indeed confusing with these various cultures, however, we do not see the need to elaborate on this, as it would take up a lot of space.

Reviewer comment: p.3 lines 10-16: I found this description confusing (assuming that I was unfamiliar with the site and the project, and could not read your ref.15, which is in Danish); much of this information is probably redundant for the purpose of this paper.

Reviewer comment: We disagree. A site description is in our opinion never redundant, as it serves to familiarise the reader with the site.

Reviewer comment: p.4 line 28: "They were then vortexed" – laboratory colloquialism!

Action taken: Changed to "Samples were then agitated using a vortex mixer"

Reviewer comment: p.5 line 26: "all proteins available for red deer were downloaded" – another colloquialism; data were downloaded or uploaded, not proteins.

Action taken: This has been amended to read " all protein sequences available for *Cervus elaphus* were downloaded "

Reviewer comment: “searched”: there are several instances (e.g. in “The Thermo RAW files generated were then searched using the software MaxQuant”) in which this term (or similar expressions such as “peak picked”) are used colloquially or ungrammatically. I think you mean that you used software X to look for reference spectra closely matching the data files.

Action taken: We understand the reviewer’s concern, but we are using common phrases in the field in this case. The code behind MaxQuant, Andromeda, is defined as a search engine, and the creators of the code also use “search” in this fashion (see Cox et al 2011 Andromeda: A Peptide Search Engine Integrated into the MaxQuant Environment <https://pubs.acs.org/doi/abs/10.1021/pr101065j>)

Reviewer comment: p.5 line 48: “Groupings were generated by applying kmeans clustering (number of clusters = 3)” – Hyphenate k-means. Groupings of what? Why 3 clusters?

Action taken: We have, based on both yours and reviewer 2’s suggestion, omitted the String network, as it was merely intended to be a way of visualising the identified proteins, but unfortunately seems to confuse readers.

Reviewer comment: p.8, line 45ff: These sentences “However, the most likely candidate, based on the ZooMS data, would be red deer since elk was part of the depauperate fauna (39). Red deer were still abundant in Denmark during the Atlantic period, as opposed to elk, which disappeared following the Littorina transgressions that turned Denmark into an archipelago.” should be re-written, because (1) the argument is not based on the ZooMS results, but on biogeography; (2) “depauperate” is not a common word (“impoverished” would be more widely understood) and the implicit point is that elk was not part of the impoverished Danish fauna; (3) the Littorina transgression is not dated; (4) the Atlantic period is not dated (I know what you mean, but many readers will not, and might assume that the Atlantic period was initiated by the Littorina transgression). A much simpler chain of reasoning might be that there is abundant red deer bone at Syltholm but no elk bones have been identified, if you know this to be the case.

Action taken: We thank the reviewer for this suggestion to better improve the readability and accuracy of this statement. It was meant as the best way to interpret the ring based on ZooMS data and site information alone, without LC-MS, but it appears to not fulfill its intention. It has been changed to: “However, given the choice between the two species indicated by ZooMS, *Cervus* would be the most likely candidate. *Alces* disappeared at some point in time in this area due to rising sea-levels (Littorina transgressions (starting c. 8,400 BP)) that effectively turned Denmark into an archipelago, whereas *Cervus* was still abundant in Denmark (40) at that time. *Alces* material could have been introduced to the site through trade, however no evidence of this species has been recovered at Syltholm from this period.”

Reviewer comment: Table 1 and Figure 2: what is a “fishing ruse”?

Action taken: We thank the reviewer for spotting this mistake. It now reads “fish trap”.

Reviewer: 2

Comments to the Author(s)

In this manuscript, Jensen et al. attempt to determine the species and skeletal element of origin of an osseous ring from Denmark. They perform μ CT, ZooMS, and LC-MS to evaluate it. This is an interesting study, but has some major interpretation issues that make its conclusions unconvincing. Aspects of the experimental setup also limit the conclusions as well.

Reviewer comment: One minor thing to revise throughout, for clarity throughout please refer to red deer and elk by their scientific names only. Elk has a very different meaning in North America (where they are called moose) than it does in Europe. Additionally, elk in North America is *Cervus elaphus canadensis* so it can be extra confusing for red deer in Europe.

Action taken: This is a valid point. The decision to use non-scientific names was done for the ease of reading, and the names used was based on the majority of authors being European. We defined these terms in the beginning of the paper, but can see that this can still be confusing. Therefore, we have changed the two species to their scientific names, often shortened to the genus name for ease of reading, explained in the introduction pg (3): two species of cervids *Cervus elaphus* (red deer, hereafter referred to as *Cervus*) and *Alces alces* (European elk or North American moose, hereafter referred to as *Alces*).

Reviewer comment: Additionally, SI Table 5 and SI 6 were unavailable. I was only able to see SI 1-4, please provide them.

Action taken: We thank the reviewer for spotting this mistake. "SI Table 5" is actually referring to SI Table 3 and "SI 6" means SI 4. We have amended this in the manuscript.

Reviewer comment: Page 2 Line 56: "ZooMS is the method of choice..." I disagree that ZooMS is "the" method of choice for species ID. LC-MS provides species information and depth of characterization at the same time and can utilize proteins other than collagen I for species ID. The following sentence also refutes this one.

Action taken: We understand the reviewer's point and agree this sentence can be misleading. It has been changed to "ZooMS is often chosen for archaeological research because it can provide a rapid, cost effective species ID for samples containing collagen (i.e. bone, antler and skin) (13,14)." (pg 2)

Reviewer comment: An additional point, there is no such thing as "collagen," all collagens must be denoted by their type (e.g., collagen I).

Authors' response: We thank the reviewer for highlighting this issue. In this case we are denoting collagen containing objects in general, which would contain more than one collagen

type, which we feel do not need to be specified and listed in this context. However, we have checked the other instances where “collagen” is used to ensure it is being used correctly.

Reviewer comment: Page 3 Line 22-24: Was the ring found in the brown gyttja layer? Was the layer it was found in reworked? If so, please clarify here.

Authors' response: The ring was indeed found in the brown gyttja. The gyttja represents a natural build up of organic matter. Whether it was “reworked” we do not know, as there is not apparent stratigraphy.

Reviewer comment: Page 3 Line 52: Should pixel be voxel for this CT data? Usually each data point in CT data is a 3D point (i.e., a voxel) and not a 2D point (pixel).

Authors' response: The voxel size (3D) is given by the efficient pixel size on the detector system, which depend on the physical pitsize in the detector and the geometric and optical magnification. The measurement in a micro-CT is a series of 2D images which is transformed into a 3D volume by the mathematical reconstruct. Therefore the acquired data is a 2D stack and the element is a pixel.

Reviewer comment: Page 3 Line 55-56: “The resulting 3D volumes are cylinders with a diameter and height of 3.2 cm and 1.35 cm respectively corresponding to the different pixel resolution.” This does not make sense to me. Please clarify how differences in voxel size result in different cylinders of the same object.

Authors' response: The camera records the images as 1024 pixel by 2014 pixels (2D array), in this study two different efficient pixelsizes has been used; 32.3 μm and 13.5 μm . As a result each 2D image has a field of view of roughly 1000 times the pixel size (1024 times pixelsize). As a result the reconstructed 3D volumes becomes cylinders of 3.2cm and 1.35cm in diameter and height. The size of the cylinders have nothing to do with the size of the object, in this case the large cylinder contains more of the object than the small cylinder. But the small cylinder has a higher resolution enabling smaller features to be seen. This is just like changing the magnification on an optical microscope.

Action taken: We have changed the text to; The resulting 3D volumes are cylinders with a diameter and height of 3.2 cm and 1.35 cm respectively corresponding to the different pixel resolutions, **containing different amounts of the object"** (pg 4)

Reviewer comment: Page 3 Line 57-60: Why were these data not analyzed using a more standard approach to CT data? There are open access options (e.g., imageJ and boneJ) that can open voxel data and allow for analysis. ImageJ also allows for segmentation of the data and 3D visualization is possible as well. Please re-analyze the data with one of these tools.

Action taken: We have redone the analyses, and the text now reads; **Visualisation was performed using ‘Avizo 9.7’ (Thermo Fisher Scientific). The volume investigated with high**

resolution X-ray micro CT, has been segmented into elements of the bone (shown in transparent gray) and porosity in the bone (shown in blue). The different levels of blue is a result of the amount of transparency. (pg 4)

We have added a new reconstructed image (Fig. 3) based on the above, and changed the figure caption to: E) network of osteons arranged longitudinally and Volkmann's canals aligned perpendicular to the latter (in blue). (pg 21)

Reviewer comment: Page 4 Line 17-20: "These samples were taken from a specimen that was defleshed by heating it in water for three days at c. 65°C. This defleshing process means that it has an equivalent thermal age (22) of around 2.5 kya at Syltholm, (assuming an activation energy of 173kJ mol⁻¹ for collagen degradation at the site whilst buried in sediment on the seafloor at a mean annual temperature of 7.5°C)," Please remove reference to thermal age. This sample is better described as an experimentally heated extant sample for comparison. Additionally, 65C is not going to drastically change this sample over 3 days, so I don't see how it could reflect 2.5 kya of aging.

Action taken: The text has been changed to "These samples were taken from a specimen that was defleshed by heating it in water for three days at c. 65°C. This experimentally heated extant sample is more comparable to the ring sample than fresh bone or antler. (pg 4)

Reviewer comment: Page 4 Line 26-36: Why were the first supernatants discarded? Schroeter et al. 2016 has shown clearly that discarding supernatants during bone protein extraction is ill advised because it leads to a loss of information. Schroeter, E. R.; DeHart, C. J.; Schweitzer, M. H.; Thomas, P. M.; Kelleher, N. L., Bone protein extractomics: comparing the efficiency of bone protein extractions of Gallus gallus in tandem mass spectrometry, with an eye towards paleoproteomics. PeerJ 2016, 4, e2603.

Authors' response: This stage of the extraction protocol is in effect a wash step prior to the extraction proper. The supernatants were therefore discarded as part of an attempt to reduce contamination from the burial environment. We thank the reviewer for reminding us of Schroeter et al. 2016. However, it is pertinent to point out that the supernatants mentioned by the reviewer are not equivalent to the "demineralization fractions" discussed by Schroeter et al. 2016.

Action taken: To make this more clear we have amended the text as follows: "The samples were incubated in 100 µL of 50 mM NH₄HCO₃ (Sigma) for 16 hours at ambient temperature. Samples were then agitated using a vortex mixer for 15 seconds before centrifugation at 13,000 rpm for 1 min, the supernatant was discarded. This step acts as a wash to limit contamination from the burial environment. After, two different extractions were performed for the ring (Extraction 1 and 2)," (pg 4)

Reviewer comment: Additionally, what were the protein concentrations after extraction? This concentration impacts the ratio of trypsin to protein and the resultant digestion completeness.

Authors' response: We agree with the reviewer that this is can be an important step in maximising the efficiency of digestion and recovery of proteomic data. However, the current protocol for ZooMS preparation (based on van Doorn, N. L., Hollund, H., & Collins, M. J. (2011). A novel and non-destructive approach for ZooMS analysis: ammonium bicarbonate buffer extraction. *Archaeological and Anthropological Sciences*, 3, 281–289.) uses a set amount of enzyme, not based on protein concentration. While we can see that this protocol is not optimal, it is sufficient for normal ZooMS analysis and contributes to the fast, easy, and low-cost aspect of this methodology. Further LC-MS analysis was only performed due to the unfortunate inability of ZooMS to separate the two species, and the same extract was used as no more sample could be taken. However, since we performed LC-MS, we wanted to also capitalise on the fact that it provides additional protein information, and not waste that potential, and hence why we also examine and try to interpret the other proteins present, although we are aware it was not prepared with an optimal protocol. We counter for the fact that digestion may not have been complete by including an allowed 2 missed cleavages in our search.

Reviewer comment: Page 5 Line 18-20: Please describe all of the mass spectrometry parameters that were used here as well (In short form is fine).

Action taken: The details were originally only referenced for ease of reading, but for those that are interested we have now included them in the method section “They were then vacuum centrifuged and resuspended as above, with 5 μ L of sample analysed by LC-MS/MS. The LC-MS/MS parameters were the same as previously used for palaeoproteomic samples (25), in short: MS1: 120k resolution, maximum injection time (IT) 25 ms, scan target 3E6. MS2: 60k resolution, top 10 mode, maximum IT 118 ms, minimum scan target 3E3, normalised collision energy of 28, dynamic exclusion 20 s, and isolation window of 1.2 m/z.” (pg 5)

Reviewer comment: Page 5 Line 23-26: Why was *Bos taurus* used to fill in missing data on Cervid data instead of the more closely related and available white-tailed deer collagen sequences from the white-tailed deer genome?

Authors' response: In this case we didn't want to bias the results to either *Cervus elaphus* or *Alces alces* by including a species that appears to be more related to *Alces* (based on Maximum-likelihood tree provided by Immel et al. 2015, <https://doi.org/10.1038/srep10853>). Those portions of the sequence are unknown and were filled in only to help with data analysis, as MaxQuant works better with complete sequences. The actual sequences in there don't really matter, as they would be the same in both species in this case (because we made them the same) and cannot be used for species ID. We decided to use *Bos taurus* specifically because the sequence is not just based on genomic transcription and is very well known, and would still be very similar to both species of interest.

Reviewer comment: Page 5 Line 42-49: String networks are not particularly meaningful here because basically all of the detected proteins are derived from the ECM. They also

mis-split the protein groups because bone has many of the serum proteins as part of the ECM. I would recommend removing them and only including the Venn diagram.

Action taken: We have, based on both yours and reviewer 1's suggestion, omitted the String network, as it was merely intended to be a way of visualising the identified proteins, but unfortunately seems to confuse readers.

Reviewer comment: Page 5 Line 59-60: Unless you have additional context for stable isotope species restriction, you can't say anything other than that this is a C3 browser. Please revise.

Authors' response: We understand the reviewer's concern, and realise that a misunderstanding has occurred. The harpoon is known to be roe deer, and we are simply saying that its values are consistent with other published roe deer data. We are not trying to identify the harpoon from the stable isotope values.

Action taken: We have changed the sentence to say "The collagen yield of the harpoon sample was 5.1% and it yielded stable isotope values ($\delta^{13}\text{C} = -24.1\text{‰}$, $\delta^{15}\text{N} = 4.4\text{‰}$ and $\text{C/N} = 3.3$) consistent with already published archaeological roe deer values in Denmark (31)." to hopefully make it a bit more clear. (pg 6)

Reviewer comment: Page 6 Line 3-15: Is there any evidence of mixing in this layer? It was unclear in the stratigraphic description. If there is mixing then a safer restricted space is likely the full age range detected here. If there is more evidence to conclude they are similar age, please add it. If there is no evidence of mixing, then the as written age makes sense.

Authors' response: The ring was indeed found in the brown gyttja. The gyttja represents a natural build up of organic matter. Whether it was "mixed" we do not know, as there is not apparent stratigraphy.

Reviewer comment: Page 6 Line 19-25: How can you tell what are primary and secondary osteons from these uCT data? These data do not have the resolution do this. Histological analysis would likely be better in this case (but may not be possible with sampling restrictions). Additionally, please add the normal diameter for deer bone vasculature (only antler was reported).

Action taken: As the reviewer correctly pointed out - histological analysis would provide greater resolution but is precluded by sampling restrictions. Text has been updated: "The diameter of osteons in antler ranges from 100–225 μm diameter, similar to those found in bovine femur (33)" See figure 4 from the referenced publication. (pg 6)

Reviewer comment: Page 6 Line 47-48: "Given the spectral similarity between the two extractions, it is likely that the artefact was not contaminated with humic acids from the

environment” If humics were present, what differences would you expect to see? What color were the extractions? Is it possible the humic substances are generating ion suppression reducing the intensity of your Extraction 1 sample that you are recovering in Extraction 2?

Authors' response: Interesting points. If humics were present we would expect Extraction 2 to be richer than Extraction 1, due to the removal of their interference. Both extracts were quite colorless - a very slight tinge of yellow when observed against a white background. The images of the artefact presented in the manuscript attest to the lack of humic contamination. We are of the opinion that ion suppression is not a dominant factor in the minimal differences between the two extractions.

Reviewer comment: Page 7 Line 5: Why is collagen I alpha 1 and alpha 2 combined? That is atypical for data searching. Also was the searched sequence only the mature collagen I form or does it contain the propeptides?

Authors' response: This is standard practice in our laboratory because the primary purpose of sequence analysis is species identification. Functionally, we consider collagen at the level of the quaternary structure, thus both chains are part of the same protein. Typically Col1A2 is more variable than Col1A1, but there are useful markers in both sequences. Combining them means that there is one top hit concatenated Collagen 1 which reflects the majority probability based on sequence information from both sequences.

The COL1 sequences do not include propeptides. This is common practice for example see Welker et al. 2016.

Reviewer comment: Page 7 Line 8-9: “blood proteins (alpha 2-HS glycoprotein (AHSG), pigment epithelium-derived factor/ serpin family F member 1 (SERPINF1), immunoglobulin gamma-1 heavy chain (IGHG1), serum albumin (ALB), apolipoprotein A-I (APOA1), and nucleobindin 1 (NUCB1))” AHSG is associated with mineral and PEDF is secreted by osteoblasts, so aren't really blood proteins in this case. Please revise.

Action taken: This has been revised to include them in the extracellular matrix proteins.

Reviewer comment: Page 7 Line 54: “However, the plasma protein APOA1 was found to be uniquely shared between the ring and the heated modern red deer antler” APOA1 is not uncommon in other bone studies, so it is best to not assume it is specific for antler. (e.g., Schroeter, E. R.; DeHart, C. J.; Schweitzer, M. H.; Thomas, P. M.; Kelleher, N. L., Bone protein extractomics: comparing the efficiency of bone protein extractions of Gallus gallus in tandem mass spectrometry, with an eye towards paleoproteomics. PeerJ 2016, 4, e2603., Cleland, T. P., Human Bone Paleoproteomics Utilizing the Single-Pot, Solid-Phase-Enhanced Sample Preparation Method to Maximize Detected Proteins and Reduce Humics. Journal of Proteome Research 2018, 17, (11), 3976-3983., Sawafuji, R.; Cappellini, E.; Nagaoka, T.; Fotakis, A. K.; Jersie-Christensen, R. R.; Olsen, J. V.; Hirata, K.; Ueda, S., Proteomic profiling of archaeological human bone. Royal Society Open Science 2017, 4, (6), 161004.)

Authors' response: We are aware that APOA1 is found in other bone studies, and is not specific to antler. At this point in the results section we are simply stating that it is the only protein uniquely recovered between the antler and the ring, and (the sentence after) that there were no uniquely recovered proteins between the ring and the bone.

Action taken: However, in regards to the reviewer's concerns, we have slightly reworded the offending sentence (slightly moved due to the removal of the STRING analysis) to specify that it is just a statement of results of our analysis (pg 8): "The plasma protein APOA1 was uniquely recovered in our extractions of the ring and the heated modern *Cervus* antler, this is also visualised in a Venn-diagram (Fig. 6). No unique proteins were recovered between only the ring and heated modern *Cervus* bone." We further adjusted our comment on this observation in the discussion (pg 10): "We also discovered multiple blood proteins in the ring (IGHG1, ALB, APOA1 and NUCB1). Growing antlers are a highly vascular tissue and contain at least twice as much blood at their peak growth as ovine rib bones, which decreases as the antler ossifies (56). APOA1, a major component of plasma high density lipoprotein shown to be linked to osteoblastogenesis and bone synthesis (Papachristou and Blair 2016), was the only protein uniquely recovered between the modern antler and the ring, although it has also been found in archaeological bone (Cleland 2018; Sawafuji et al. 2017) and could represent missed recovery in the bone sample. It is of note that APOA1 was only detected by Sawafuji et al. (2017) in the infant remains studied (59), indicating association with bone formation." Further, we have omitted the STRING network, so Fig. 6d is now just Fig. 6.

Reviewer comment: Page 8 Line 9-10: "This process occurs naturally over time, and while accelerated by factors such as temperature (38), these results indicate that the proteins examined are authentic due to their greater amount of damage compared to modern contamination" Deamidation is more complex than just a temperature related change (see Schroeter, E. R.; Cleland, T. P., Glutamine deamidation: an indicator of antiquity, or preservational quality? Rapid Communications in Mass Spectrometry 2016, 30, 251-255.) please clarify this here.

Authors' response: We agree with the reviewer that deamidation is a more complicated process than temperature. We only use it to indicate likelihood of authentic protein recovery, to show that the proteins we use in the analysis are likely not from modern contamination, and thus did not go deep into this discussion. Authentication using deamidation has been used in previous publications, it is beyond the scope of this paper to review deamidation chemistry. Independent of factors that accelerate deamidation, the point is that known contaminants i.e. human keratins are less deamidated than the cervid collagen.

Action taken: However, to make this point more clear, we have changed this section to read: "This damage process occurs naturally over time, and while confounded by chemical and environmental factors (such as pH, temperature, and humidity (Robinson and Robinson 2004; Schroeter and Cleland 2016)), these results indicate that the proteins examined in our analysis are likely not preserved well enough to be modern contamination, due to the

observed greater amount of damage detected when compared to known contaminant proteins.”

Reviewer comment: Page 9 Line 7-10: “For red deer, up to 30 kg of antler can be generated within 3 months (45), increasing the probability of recovering bone formation and mineralisation proteins from this tissue, compared to mature long bone” Unless there is evidence for up-regulation in these proteins in extant studies, this sentence just isn't correct. All of these proteins are routinely detected in bone studies, so detection probability has nothing to do with antler/bone.

Action taken: This sentence has been removed in the rewrite of the '*Antler or bone?*' Section (from pg 9).

Reviewer comment: Page 9 Line 10-13: “In addition, THBS1 and COL1, along with fibrillin-1, are required for the permanence of matrix scaffold construction in the initial extracellular osseous matrix formation (46), therefore, the presence of THBS1 is suggestive of maturing bony matrix at an early stage of development.” THBS1 is also associated with bone homeostasis (<https://asbmr.onlinelibrary.wiley.com/doi/abs/10.1002/jbmr.2308>) so does not strictly relate to early development as described.

Action taken: This sentence has been removed in the rewrite of the '*Antler or bone?*' Section (from pg 9).

Reviewer comment: Page 9 Line 16: If you haven't quantified how much Col11 and 12 is present then their presence here really does not support an antler origin because bone has both proteins as well.

Authors' response: Unfortunately, due to the restrictions of the ZooMS methodology, no quantification was done in this analysis.

Action taken: Label-free quantification (LFQ) was attempted, using Perseus (an add-on to the MaxQuant software) for hierarchical clustering, comparing the samples presented to other known-tissue samples from Syltholm (not included in publication), modern roe deer (*Capreolus capreolus*) bone and antler (also not included), and recently published modern antler samples (Su et al. 2019, doi: [10.7717/peerj.7299](https://doi.org/10.7717/peerj.7299); LFQ was attempted before submitting, but this paper was published afterwards and added in this second attempt). However, due to the relatively lower amount of proteins detected, and subsequent missing values, no convincing identification could be made, although the ring does cluster with an antler sample from the same site. This is provided below for reviewer:

Figure: Clustering of bone and antler samples by LFQ intensity (log2). Proteins were filtered for ≥ 2 peptides; removed known contaminants, only identified by site and reverse hits; and filtered to valid values in $>50\%$ of samples. Clustering was with Pearson correlation, average linkage, 300 clusters with 10 iterations.

Unfortunately, doing an in-depth quantitative study was beyond the scope of this research, although we highly encourage this avenue of inquiry in the future. We do not mean to say that the presence of specific proteins are 'markers' for one tissue or the other, only that these are congruent with our hypothesis based on the ease of manufacture. We agree the tissue ID is not resolved at this time. Even though these are not quantitative results, it is reasonable to assume that proteins that are more abundant in a tissue would be more likely to be recovered, especially with a non-optimised methodology and a degraded proteome.

That is why we include these proteins in our discussion, as we are discussing the possible meaning of their detection. However, we see that our point was not fully realised in the text, therefore we have added the following texts to the discussion: “Unfortunately, comparison of the ring proteome versus the antler and bone did not enable us to confidently assign the ring to either. Due to the limitations of this study, we could not perform a quantitative analysis, partly due to methodology and partly due to the limited recovery of proteins of interest. To our knowledge, there has been no quantitative proteomic comparison between these two tissues, neither with modern nor archaeological samples.” (pg 9); “Whilst our analyses were not quantitative, it is not out of the question that more abundant proteins would be more likely to be recovered, especially from a proteome depleted sample (due to taphonomic processes).” (pg 10); and “COL11 and COL12 have been recovered from archaeological bone (ex. (53) and (54), but Sawafuji et al (2017) have shown that the protein abundance score of COL12 decreases in older human individuals compared to those younger, corresponding with the relative amount of bone growth” (pg 10)

Reviewer comment: Page 9 Line 21-22: “identification is also supported by the presence of SERPINF1, APOA1, POSTN, and THBS1, all of which have been implicated in axon/nerve growth in growing antlers” None of these proteins are unique to antler (including in the literature) and are all consistently detected from bone samples. PEDF is expressed by osteoblasts, APOA1 is from blood (see citations above), periostin is produced by osteoblasts, and thrombospondin 1 is important for bone homeostasis. Please revise.

Authors’ response: We did not mean to imply that these proteins are not present in bone, just stating their use in antlers.

Action taken: We have revised this portion to say “It [APOA1], along with SERPINF1, POSTN, and THBS1, has been implicated in axon/nerve growth in growing antlers (54). SERPINF1 and THBS1 have also been implicated in the stimulation and remodelling of vasculature in antler cartilage, respectively (54,55). However, these proteins are also recovered from bone samples, albeit highly associated with bone growth and remodeling; SERPINF1 being expressed by osteoblasts during endochondrial bone formation (Quan et al. 2005), POSTIN is highly expressed in the periosteum and highly active during bone growth and contributes to changes in bone diameter and cortical thickness (Merle and Garnero 2012), and THBS1 is related to regulating bone remodeling, maintenance of bone mass, and fracture healing (Delany and Hankenson 2009; Taylor et al. 2009).” (pg 10)

Reviewer comment: Page 9 Line 27-29: “An archaeological study of human remains from different biological ages only discovered APOA1 in infants (59), indicating its preferential presence archaeologically in plasma rich, developing osseous tissue” Cleland et al. 2018 found APOA1 in adult human individuals so age does not really have much to do with APOA1 presence or absence (Cleland, T. P., Human Bone Paleoproteomics Utilizing the Single-Pot, Solid-Phase-Enhanced Sample Preparation Method to Maximize Detected Proteins and Reduce Humics. Journal of Proteome Research 2018, 17, (11), 3976-3983.)

Authors' response: We mention that, in a comparison study of archaeological individuals of different ages (REF 59), that APOA1 was age correlated, which is in agreement with other studies. We do not say that APOA1 cannot be found in bone, just that it seems to be more associated with bone/antler formation.

Action taken: We have amended the text as follows (pg 10): “We also discovered multiple blood proteins in the ring (IGHG1, ALB, APOA1 and NUCB1). Growing antlers are a highly vascular tissue and contain at least twice as much blood at their peak growth as ovine rib bones, which decreases as the antler ossifies (56). APOA1, a major component of plasma high density lipoprotein shown to be linked to osteoblastogenesis and bone synthesis (Papachristou and Blair 2016), was the only protein uniquely recovered between the modern antler and the ring, although it has also been found in archaeological bone (Cleland 2018; Sawafuji et al. 2017) and could represent missed recovery in the bone sample. It is of note that APOA1 was only detected by Sawafuji et al. (2017) in the infant remains studied (59), indicating association with bone formation.”

Reviewer comment: Page 9 Line 33-36: “the genes of three of the top five proteins with the most peptides recovered, collagen types 1 and 3 and osteocalcin, have been shown to be expressed in the ossified part of velvet antler 10-30 times more than in the skeleton of the animal” To support quantitative statements from the literature, quantification of protein amounts are necessary here as well. Qualitative detection of typically abundant proteins in bone \neq quantitative assessments of how much is present.

Authors' response: We are aware that our work is not quantitative, see above. However, we still believe that although these are not quantitative results, it is reasonable to assume that proteins that are more abundant in a tissue would be more likely to be recovered.

Action taken: We express this notion in the revised discussion as part of the line of conjecture, and still state that this is not enough evidence to distinguish between antler and bone: (Pg 9-10) “Unfortunately, comparison of the ring proteome versus the antler and bone did not enable us to confidently assign the ring to either. Due to the limitations of this study, we could not perform a quantitative analysis, partly due to methodology and partly due to the limited recovery of proteins of interest. To our knowledge, there has been no quantitative proteomic comparison between these two tissues, neither with modern nor archaeological samples (Stéger et al. 2010). Stéger et al. 2010, however, did quantitatively examine modern *Cervus elaphus* antler and bone for differences in gene expression. They found that the expression of eight genes were 10-30 fold times more expressed in the ossified portion of antler than in skeletal bone from the same individual. Four of these proteins (COL1, COL3, BGLAP, and SPARC) were also found in the ring sample. **These are all proteins associated with skeletal development, and can be found in both antler and bone.** However, it is unsurprising that bone development proteins are more abundant in antler as it is the fastest growing osseous mammalian tissue (Sadighi et al. 2001) due to yearly regeneration. Whilst our analyses were not quantitative, it is not out of the question that more abundant proteins

would be more likely to be recovered, especially from a proteome depleted sample (due to taphonomic processes).” (pg 10)

Reviewer comment: Page 9 Line 39: High sequence coverage of collagen I from limited starting bone material is not surprising and is fairly typical for recent LC-MS studies. Please revise.

Action taken: This sentence has been removed in the rewrite of the ‘*Antler or bone?*’ Section (from pg 9).

Reviewer comment: Page 9 Line 46: “heating in water - a commonly used surrogate for diagenesis (66)” Cooked bone is very different from 65C heating for 3 days. Please do not make this comparison.

Action taken: The text has been changed to “To this end we selected a *Cervus* specimen that had been experimentally heated and performed the same extraction protocol that was used for the ring sample.” (pg 9)

Reviewer comment: Page 9 Line 48: “As expected the use of thermally degraded samples yielded proteomes of similar size to that of the ring” The proteome sizes detected here are likely more an expression of the extractome of this extraction method than anything about diagenesis. Please revise.

Action taken: We agree this could be a factor and this sentence has been removed in the rewrite of the ‘*Antler or bone?*’ Section (from pg 9).

Reviewer comment: Page 10 Line 3: “tentative evidence that the Syltholm ring was manufactured from red deer antler” I’m not at all convinced that this ring came from antler and not bone. Using only qualitative data, none of the proteins detected from the ring can directly distinguish between the two bone tissues. Please revise accordingly.

Authors’ response: We did not mean to suggest with 100% confidence that the ring is made of antler tissue (hence the use of “tentative”), just that it is consistent with the manufacturing process that would make the most sense.

Action taken: But we realise that this can be read too strongly, hence we have revised this paragraph to the following: “Therefore, we suggest that the proteins recovered, especially those related to increased bone growth and high vascularisation, are consistent with a possible tissue identification of antler, suggested based on the ease of manufacture of this item from antler, however, there is not sufficient evidence to rule out bone either. We greatly encourage that more research should be undertaken to confirm the proteomic differences between antler and skeletal bone, as it would be valuable for future palaeoproteomic studies and the understanding of archaeological worked ossified objects.” (pg 10)

Reviewer comment: Page 10 Lines 27-31: Again, because I'm not convinced that this ring is derived from antler, hypothesizing manufacture is premature. Please revise.

Authors' response: The proposed chaîne opératoire here is not based on the proteomic results. It is based on the ease of manufacture of this ring from antler, compared to bone. But we realise that we have only specified antler manufacture (although it is roughly similar to bone) and we mostly mean to show that it is different from the use of tooth raw material in the only coeval ring from Denmark.

Action taken: We have amended the text as follows (pg 11): “However, one can imagine that production of such a ring as the one we studied would require: A) obtaining an antler **or bone** of a *Cervus*, B) transverse sawing to obtain a rough-out, C) scraping off the velvet bone for antler; **more intensive circular shaping for bone**, D) removal of the interior trabecular tissue using a flint borer and E) polishing of the exterior.” (pg 11)